# Partially Personalized Federated Learning: Breaking the Curse of Data Heterogeneity

**Konstantin Mishchenko** *konsta.mish@gmail.com*
*Meta, France*[*]

**Rustem Islamov**
*University of Basel, Switzerland*[†]

**Eduard Gorbunov**
*Mohamed bin Zayed University of Artificial Intelligence, United Arab Emirates*

**Samuel Horváth**
*Mohamed bin Zayed University of Artificial Intelligence, United Arab Emirates*

**Reviewed on OpenReview:** *https://openreview.net/forum?id=8tMMCf4YYn*

## Abstract

We consider a partially personalized formulation of Federated Learning (FL) that strikes a balance between the flexibility of personalization and cooperativeness of global training. In our framework, we split the variables into global parameters, which are shared across all clients, and individual local parameters, which are kept private. We prove that under the right split of parameters, it is possible to find global parameters that allow each client to fit their data perfectly, and refer to the obtained problem as *overpersonalized*. For instance, the shared global parameters can be used to learn good data representations, whereas the personalized layers are fine-tuned for a specific client. Moreover, we present a simple algorithm for the partially personalized formulation that offers significant benefits to all clients. In particular, it breaks the curse of data heterogeneity in several settings, such as training with local steps, asynchronous training, and Byzantine-robust training.

## 1 Introduction

Federated Learning has emerged as a promising approach to address data privacy concerns and enable distributed learning in various applications, such as healthcare, finance, and mobile devices (Konečný et al., 2016). In this approach, multiple parties collaboratively train a machine learning model without sharing their raw data with each other, instead only sharing the model updates with a central server. Despite its potential, Federated Learning faces significant challenges in optimizing the models due to the heterogeneity of the client's data, the heterogeneity of the devices involved in training, and the communication constraints (McMahan et al., 2017).

Collaborative Federated Learning aims at uniting a number of private data holders in order to find together a joint set of parameters $x^*$ that minimizes their private loss functions:

$$\text{find } x^* \text{ s.t. } f_m(x^*) \approx \arg\min_x f_m(x) \text{ for all } m. \qquad \text{(Non-personalized FL)}$$

This formulation, however, is often too restrictive as there might not exist a common model $x^*$ that fits all participating clients. FL, therefore, often relies on private *fine-tuning* or *personalization* to find individual

---

[*]This work was done when Konstantin Mishchenko was at Samsung AI Center.
[†]This work was done when Rustem Islamov was a Master's student at Institut Polytechnique de Paris.

$x_m^*$ for every client $m$. This leads to a fully personalized formulation of FL:

$$\text{find } x_1^*, x_2^*, \ldots \text{ s.t. } f_m(x_m^*) \approx \arg\min_x f_m(x) \text{ for all } m.$$

(Personalized FL)

This formulation offers a lot of flexibility at the cost of needing to do more work for every client. Since there is no longer a global model $x$ trained to perform well on all clients, it is now necessary to find a personalized model for every new client. New clients, therefore, might experience bad performance of the initial model until they obtain enough data to personalize it for their needs.

Thus, having no personalization might be too restrictive, while full personalization can be excessively expensive and inefficient. Following recent works on personalization by Arivazhagan et al. (2019) and Singhal et al. (2021), we strike a balance by splitting the parameters into *global* and *private* parts and formulate our problem as

$$\text{find } \theta^*, w_1^*, w_2^*, \ldots \text{ s.t. } f_m(\theta^*, w_m^*) \approx \arg\min_{\theta, w} f_m(\theta, w)$$

for all $m$. (Partially personalized FL)

Above, $\theta^*$ is the part of the parameters that is shared by all clients, while parameters in $w_m^*$ are specific to client $m$.

## 1.1 Motivation and formulation

The main interest of our work is to show that the partially personalized formulation gives a significant benefit to all clients as long as there exists a universal $\theta^*$. As a motivating example, consider the phenomenon in deep learning where we can train a big network on one task or dataset and fine-tune only its last few layers when working with new source of data. The layers that are not fine-tuned produce so-called *representations*. The overall objective is then to minimize $\ell(\Phi(\theta, \mathcal{X}_m), w_m, y_m)$, where $\ell(\cdot, \cdot, \cdot)$ is a loss function, $\Phi(\theta, \cdot)$ is the features mapping parameterized by $\theta$, $\mathcal{X}_m, y_m$ are the data of the $m$-th client, and $w_m$ are fine-tuning parameters. Since the representation should fit all non-adversarial clients and we send to clients only $\theta$, we expect the $m$-th client to find approximately $w_m^*(\theta) = \text{argmin}_w f_m(\theta, w)$. Therefore, we want to find such $\theta$ that changing only $w_m$ is sufficient, which can be written as $\nabla_1 f_m(\theta, w_m^*(\theta)) = 0$, where $\nabla_1 f_m$ is the gradient of $f_m$ with respect to $\theta$ that does not differentiate through $w_m^*(\theta)$. Thus, our overall objective is to solve

$$\text{find } \theta^* \in \mathbb{R}^d \text{ s.t. } F_m(\theta^*) = 0 \text{ for all } m,$$
$$\text{where } F_m(\theta) = \nabla_1 f_m(\theta, w_m^*(\theta)). \tag{1}$$

The values of private $w_m$'s are not available to the server to preserve the privacy of the clients. Moreover, to enable *stateless* FL applications where a client cannot maintain a state, we assume that a client might initialize its $w_m$ randomly. To the best of our knowledge, this exact problem has not been studied in optimization literature. It can also be seen as a first-order stationarity condition for the overparameterized objective

$$\min_{\theta, \{w_m\}_m} \mathbb{E}_m[f_m(\theta, w_m)],$$

which connects our work to the previous literate on overparameterization in stochastic optimization (Ma et al., 2018; Vaswani et al., 2019).

## 1.2 Our objective

Motivated by the scenario of representation learning, we ask about the potential benefits of partial personalization. In summary, the key aspects of our work are as follows:

1. The problem is partially personalized with global parameters $\theta$ (e.g., representations) and local/private parameters $w$ (e.g., last layers).

2. The data is heterogeneous, but there are sufficiently many parameters in $w$ to make the solution set of problem (1) non-empty. Similar to overparameterization in deep learning (Allen-Zhu et al., 2019), we call this setting *overpersonalized*.

3. The clients are *stateless* by default, i.e., they might not be able to store their private parameters $w$, as is often the case in cross-device FL (Kairouz et al., 2021).

4. The clients should be able to run arbitrary local optimizers to fine-tune the private part of the model.

5. Our particular interest is in identifying the benefit of local training as well as the positive impact of client cooperation.

It is natural to ask if this objective is reasonable at all. As the next proposition shows, it is always feasible if we allocate sufficiently many parameters in $w$.

**Proposition 1.** There always exists a split of parameters $x$ into global and local parameters $\theta, w_1, w_2, \ldots$ such that there exists $\theta^*$ satisfying $F_m(\theta^*) = 0$ for all $m$.

*Proof.* The result is trivial: if we set $w_m$ to be all parameters and $\theta$ to be any variable such that $f_m$ does not depend on it, then naturally $f_m(\theta, w_m^*(\theta)) = \min_{\theta', w} f_m(\theta', w)$ and $F_m(\theta) = 0$ for all $\theta$. □

As indicated by Proposition 1, splitting the variables to remove the data heterogeneity is always possible. It is non-trivial, on the other hand, to find the right split, as moving all parameters into $w$ will eliminate any cooperation between the clients. The interesting case is, therefore, when $\theta$ includes sufficiently many parameters that affect $f_m$. We leave the question of finding the right split for future work and focus instead on the consequences of having it. In particular, we prove, under some mild assumptions, that the partially personalized objective can be cast as a nonlinear equation:

$$\text{find } \theta^* \text{ s.t. } F(\theta^*) = 0, \tag{2}$$

where $F$ is a nonlinear operator. Moreover, we design algorithms capable of leveraging this property that demonstrates, for the first time, the provable benefits of partial personalization.

## 1.3 Motivating challenges

**Device heterogeneity.** So far we have been discussing how the data heterogeneity is a challenge for federated learning. At the same time, heterogeneity in the devices participating in training leads to unbalanced computation time of different clients. This issue also arose in classical distributed optimization, where it was proposed (Niu et al., 2011) to run *asynchronous* methods that do not ask the participating devices to synchronize their updates.

While asynchronous methods have been popular in practice in many applications, including Federated Learning (Nguyen et al., 2022), their theoretical utility is very limited in the setting of heterogeneous data. This is an expected drawback as when all clients have different functions, clients that participate less will not give us sufficient information, leading to a biased solution. Nevertheless, in this paper, we show that in the context of learning representations, even with heterogeneous data, Asynchronous SGD would converge under arbitrary delays.

**Byzantine attacks.** The next challenge that we will consider is that of the potential presence of *Byzantine* clients, which is a problem faced by many distributed systems (Lamport et al., 1982; Su & Vaidya, 2016; Lyu et al., 2020). Standard Federated Learning algorithms are vulnerable to Byzantine attacks even in the case of homogeneous data. For general heterogeneous problems, Byzantine robustness cannot be achieved in general. In this work, we develop a new method for the proposed problem formulation and show its provable robustness to Byzantine attacks, even in the case of heterogeneous data on clients.

Table 1: A brief overview of related work and conceptual differences to our approach. We say that *local steps are provably helpful* if the method achieves a better communication complexity when increasing the number of local steps. A method *handles heterogeneous data* if its complexity does not depend on any heterogeneity constant. In "Local Training with GD", clients do not communicate at all and run Gradient Descent using local data (non-distributed method).

| Algorithm | Personalized | Stateless | Provably helpful local steps | Handles heterogeneous data |
|---|:---:|:---:|:---:|:---:|
| Local Training with GD | ✓ | ✓ | ✓ | ✓ |
| FedAvg (McMahan et al., 2017) | ✗ | ✓ | ✓ | ✗ |
| FedProx(Li et al., 2020a) | ✗ | ✓ | ✓ | ✗ |
| Scaffnew (Mishchenko et al., 2022b) | ✗ | ✗ | ✓ | ✗ |
| FedAlt/FedSim (Pillutla et al., 2022b) | ✓ | ✗ | ✗ | ✓ |
| FFGG (**This work**) | ✓ | ✓ | ✓ | ✓ |

**Personalization of representations.** Beyond learning representations and personalizing the last layers, a common framework is to personalize representations while the weights in the last layers can be shared. For instance, in medical applications, different scanners or sensors may have different intensities and contrast, resulting in a feature shift (Li et al., 2020b). The labels, however, are shared across hospitals, so we can still train personalized models with shared parameters. In that case, $w_m$ would be responsible for aligning the features, for instance, using batch normalization parameters as done by Li et al. (2020b), whereas $\theta$ would represent the rest of the network.

Similarly, in cross-device Federated Learning, the features might change due to use of different phones, giving another motivation for personalizing the representations. As every phone model is equipped with its own version of a photo camera, the layers used to produce the embeddings may need to be readjusted, while the layers deeper in the network can remain the same. Likewise, a robot can be deployed in places with different lighting, so its visual signal may depend on the environment, suggesting that the first layers need to be adjusted. By and large, hardware and environment heterogeneity will demand changes to the part of the network giving us representations, whereas the decision layers can remain intact in such scenarios.

On a final note, personalizing the first layers, which interact with the data, might boost privacy even more than the personalization of the last layers. These layers can be used by clients to obfuscate the data source and distribution while relying on the server to get good universal predictors for generic obfuscated data. As our framework does not assume the exact meaning of $\theta$ and $w$, all of our results are immediately applicable in the setting of personalizing the representation.

## 1.4 Related work

**Methods for non-personalized FL.** Most methods for Federated Learning stem from the Federated Averaging algorithm (FedAvg) of McMahan et al. (2017). FedAvg itself can be seen as a variant of Local SGD, which received a lot of attention in the optimization literature (Mangasarian, 1995; Stich, 2019). The speed of convergence of Local SGD heavily depends on the amount of data heterogeneity, with much slower rates under arbitrary data dissimilarity (Khaled et al., 2020). Other methods, such as FedProx (Li et al., 2020a) and Scaffold (Karimireddy et al., 2020b), were proposed to alleviate the issues stemming from data heterogeneity, with provable acceleration if clients can maintain local states (Mishchenko et al., 2022b; Grudzień et al., 2022). To the best of our knowledge, however, the case of FL with stateless clients and heterogeneous data remains challenging.

**Personalization.** FedAvg and other classical algorithms train a single model for all clients. Jiang et al. (2019) showed that FedAvg is good at personalization, but this is an implicit property. Various works have proposed alternative methods that run algorithms with explicit personalization, for example using model-agnostic meta-learning (Chen et al., 2018; Fallah et al., 2020) or using penalty-based objective (Hanzely & Richtárik, 2020). Unlike us, these papers consider full-model personalization and update all parameters.

**Partial personalization.** More related to ours, some works split variables into personalized and non-personalized ones. In the context of Meta-Learning with deep networks, Raghu et al. (2020) established that only the last few layers needed to be adapted for new tasks. The work of Pillutla et al. (2022b) is perhaps the closest to ours as they studied optimization properties of partially personalized FL. In particular, they established convergence of algorithms with alternating and simultaneous updates, but unfortunately, their theory does not guarantee any benefit from using multiple gradient steps beyond reducing noise due to extra sampling.

**Asynchronous optimization.** Asynchronous SGD has been extensively studied for homogeneous data (Tsitsiklis et al., 1986; Agarwal & Duchi, 2011). Recently, it was shown that Asynchronous SGD converges regardless of the delays in the setting of homogeneous data (Mishchenko et al., 2022a; Koloskova et al., 2022). Moreover, for heterogeneous data, Asynchronous SGD was to converge to a neighborhood under gradient similarity (Mishchenko et al., 2022a) and to the exact solution under stochastic sampling of clients (Koloskova et al., 2022; Islamov et al., 2024). However, to the best of our knowledge, Asynchronous SGD is not guaranteed to converge for heterogeneous data without assuming structure in the delays.

**Byzantine robustness.** The first approaches to Byzantine-robust distributed learning were concentrated around the replacing of standard averaging in Parallel SGD by some other aggregation rule, which is more robust to the outliers (Blanchard et al., 2017; Yin et al., 2018; Damaskinos et al., 2019; El-Mhamdi et al., 2018; Pillutla et al., 2022a). However, they turned out to be vulnerable to special Byzantine attacks (Baruch et al., 2019; Xie et al., 2020) that can be partially explained by the fact that the term "robust aggregation" was formally introduced later by Karimireddy et al. (2021). In the case of homogeneous data, this discovery led to the development of provably Byzantine-robust distributed methods (Karimireddy et al., 2021; Gorbunov et al., 2022; 2023). In parallel, alternative approaches based on clients' elimination were developed in (Alistarh et al., 2018; Allen-Zhu et al., 2021).

In the arbitrary heterogeneous data case, it is impossible to distinguish regular clients from Byzantine ones. Moreover, even when the heterogeneity (in terms of the gradients' dissimilarity) is bounded by a constant, there is a lower bounded on the optimization error that can be achieved for a given fraction of Byzantine clients (Karimireddy et al., 2022), matching/nearly matching upper-bounds were derived in (Karimireddy et al., 2022; Wu et al., 2020; Zhu & Ling, 2021; Gorbunov et al., 2023). Nevertheless, in the overparameterized regime, some methods can still converge to the exact solution (asymptotically) – this was shown in (Karimireddy et al., 2022; Gorbunov et al., 2023) for standard Federated Learning formulation. In our work, we continue this line of work by showing that for the proposed problem formulation, one can also achieve provable Byzantine robustness even in the heterogeneous case under similar assumptions about overparameterization.

**Bilevel optimization.** The partially-personalized FL formulation can be seen as a special case of bilevel optimization. Recently, bilevel optimization was also studied in the context of FL (Tarzanagh et al., 2022; Li et al., 2022). Its main disadvantage is the requirement to differentiate through the inner-level objective, which is typically done by estimating inverse-Jacobian-vector products. Moreover, algorithms for bilevel optimization usually require maintaining the private variable in memory and updating over multiple communication rounds, which is not possible in stateless Federated Learning (Kairouz et al., 2021).

## 2 Technical Preliminaries

**Notation.** We say that a function is $L$-smooth if its gradient is $L$-Lipschitz. We use $\nabla_i f_m(\cdot, \cdot)$ to denote the gradient with respect to the $i$-th argument of $f_m$.

**Definition 1.** For any client $m$ and parameters $\theta$, we denote $w_m^*(\theta)$ as any minimizer of the $m$-th client loss, i.e.,

$$w_m^*(\theta) \in \arg\min_w f_m(\theta, w). \tag{3}$$

**Assumption 1.** *We assume that the operator* $F_m(\theta) = \nabla_1 f_m(\theta, w_m^*(\theta))$ *is* $\frac{1}{L}$-*cocoercive in* $\theta$, *i.e., for any* $\theta_1, \theta_2$

$$\langle F_m(\theta_1) - F_m(\theta_2), \theta_1 - \theta_2 \rangle \geq \frac{1}{L} \|F_m(\theta_1) - F_m(\theta_2)\|^2. \tag{4}$$

The best way to view Assumption 1 is as a generalization of convexity and smoothness to nonlinear operators. For instance, if $f_m$ does not depend on $w$, then it is enough for it to be convex and $L$-smooth (Nesterov, 2013). However, our main interest is when parameters $w$ are personalized, so below, we give a few examples with nontrivial dependence on $w$ where it provably holds.

Another assumption that we make is regarding the overparameterization aspect of the problem. For example, if good data representations exist for all clients, then we can fine-tune $w$ on each client to perfectly fit the training data. This is formalized below.

**Assumption 2.** *There exists at least one $\theta^*$ such that for any client $m$, the loss $f_m(\theta^*, w)$ achieves its minimum with some optimal $w_m^*(\theta^*)$, i.e., $f_m(\theta^*, w_m^*(\theta^*)) = \min_{\theta, w} f_m(\theta, w)$.*

Notice that Assumption 2 requires that $\theta$ does not represent personalization to be possible.

Now we give a few examples for which Assumption 1 holds. Note that we defer all proofs to the appendix.

**Example 1.** Let $f_m(\theta, w)$ be given as $f_m(\theta, w) = \phi_m(\theta) + \frac{1}{2}\|\mathbf{A}_m\theta + \mathbf{B}_m w - y_m\|^2$, where $\phi_m$ is any convex $L_\phi$-smooth function. Then, Assumption 1 is satisfied with $L = 2\max(L_\phi, \|\mathbf{A}_m^\top(\mathbf{I} - \mathbf{B}_m\mathbf{B}_m^\dagger)\mathbf{A}_m\|)$.

Example 1 is a regularized version of the linear regression problem, which has served as a litmus test for verifying if an assumption makes sense. Our assumption is thus validated on this simple example. At the same time, Assumption 1 is satisfied for a wider range of functions that include the following example.

**Example 2.** Let $f_m$ be twice-differentiable, $\mu$-strongly convex and $L$-smooth in $\theta$. Moreover, assume the cross-term in the Hessian and the Jacobian of $w_m^*$ to be bounded as $\|\nabla_{12}^2 f_m(\theta, w)\| \leq C_1$, and $\|\nabla_\theta w_m^*(\theta)\| \leq C_2$ with $C_1 C_2 \leq \frac{\mu}{2}$. Then, $F_m$ is cocoercive and $\frac{\mu}{2}$-strongly monotone.

Example 2 provides a general recipe for an objective to satisfy Assumption 1. It is somewhat restrictive as it requires smoothness of both $f_m$ and $w_m^*$, but these assumptions are, in fact, natural. The smoothness of $f_m$ is commonly assumed to make minimization of $f_m$ possible (Nesterov, 2013). The assumption on the smoothness of $w_m^*(\theta)$ is directly related to the assumptions employed in the theory of min-max optimization; see, for example, (Nouiehed et al., 2019). Moreover, it is possible to ensure $w_m^*$ is Lipschitz by adding $\frac{\lambda}{2}\|w\|^2$ to $f_m$, see Lemma B.1 of Nouiehed et al. (2019).

Let us also add a small generalization of Example 1 to the case of the composition of smooth nonlinear functions and linear transformations.

**Example 3.** Let $f_m(\theta, w)$ be given as $f_m(\theta, w) = \phi_m(\theta) + \psi_m(\mathbf{A}_m\theta + \mathbf{B}_m w - y_m)$, where $\psi_m$ and $\phi_m$ are any convex and $\frac{L}{2}$-smooth functions. Then, Assumption 1 is satisfied.

This example is particularly interesting from the perspective of parameter-efficient fine-tuning. Methods such as Low-Rank Adaptation (LoRA) (Hu et al., 2022), fine-tune linear layers by adding low-rank perturbations. Specifically, the linear layer is split into two parts, similar to how we compute $\mathbf{A}_m\theta + \mathbf{B}_m w$ in Example 3. Note that $\mathbf{B}_m$ can have a rank much smaller than $\mathbf{A}_m$.

## 3 New Method: Fine-tuning Followed by Global Gradient (FFGG)

Having defined a formulation of partial personalization, it is very easy to derive an algorithm that will solve it. Under our assumptions, it is enough to update the global parameters $\theta$ using the standard gradient-descent update. However, in practice, we need to compute $w_m^*(\theta)$ using local gradient updates. This gives us a double-loop method called Fine-tuning Followed by Global Gradient (FFGG), which we present in Algorithm 1. At round $r$ of FFGG, we sample a set of clients $C^r$, and the clients in the set approximately minimize their loss functions w.r.t. the personalized parameters $w$ and fixed global parameters $\theta^r$ to get $w_m^r$ (Line 4). For example, clients can perform $\tau$ local steps with stepsize $\gamma_w$ and find an approximate solution $w_m^r := w_m^{r,\tau}$ of their local objective using the gradients $\nabla_2 f_m(\cdot, \cdot)$ (see Algorithm 2). Then, the clients use $w_m^r$ to compute the gradient $\nabla_1 f_m(\cdot, \cdot)$ and send their update $\Delta_m^r$ to the server (Line 5). After that, the server averages the received vectors $\Delta_m^r$ and makes a gradient-descent type update of $\theta^r$ with stepsize $\gamma_\theta$ (Line 7). The proposed method is applicable to situations when clients are stateless since neither global parameters $\theta^r$ nor local parameters $w_m^r$ are required to be stored on client $m$ once the round is finished.

---

**Algorithm 1** Fine-tuning Followed by Global Gradient (FFGG)

---

**Input:** initialization $\theta^0 \in \mathbb{R}^d$, stepsize $\gamma_\theta > 0$

1: **for** $r = 0, 1, 2, \ldots$ **do**
2:     Sample a batch of clients $C^r$
3:     **for** client $m \in C^r$ **do**
4:         Solve $w_m^r \approx \mathrm{argmin}_w f_m(\theta^r, w)$
          (E.g., using Algorithm 2)
5:         $\Delta_m^r = \nabla_1 f_m(\theta^r, w_m^r)$
6:     **end for**
7:     $\theta^{r+1} = \theta^r - \gamma_\theta \frac{1}{|C^r|} \sum_{m \in C^r} \Delta_m^r$
8: **end for**

---

**Algorithm 2** Local GD fine-tuner
(to find $w_m^r \approx \mathrm{argmin}_w f_m(\theta^r, w)$)

---

**Input:** stepsize $\gamma_\theta > 0$, number of local steps $\tau$, $\theta^r \in \mathbb{R}^d$

1: Initialize $w_m^{r,0}$ randomly
2: **for** $i = 0, \ldots, \tau - 1$ **do**
3:     $w_m^{r,i+1} = w_m^{r,i} - \gamma_w \nabla_2 f_m(\theta^r, w_m^{r,i})$
4: **end for**

---

**Comparison with (Pillutla et al., 2022b).** We also notice that our method is noticeably different from the ones proposed in (Pillutla et al., 2022b). From the methodological perspective, Pillutla et al. (2022b) studies how gradient-based methods popular in FL solve the considered problem. Our approach, in contrast, is based on the observation that the objective is similar to bilevel optimization with the restriction that we cannot store auxiliary variables due to the stateless nature of the clients. Therefore, the method that we proposed tries to first find appropriate private parameters $w_m(\theta)$ and only then update $\theta$, whereas the FedSim method from (Pillutla et al., 2022b) updates $\theta$ on each worker without first trying to see if a client's local objective can be minimized by updating only $w_m$.

There is more similarity between our FFGG and the FedAlt method from (Pillutla et al., 2022b) with $\tau_v = 1$ (number of local steps w.r.t. $\theta$). In particular, if we use gradient descent as the local solver when optimizing $w_m$ in FFGG, the methods become almost identical. However, the code of Pillutla et al. (2022b) only supports running the same number of updated with respect to $\theta$ and $w_m$, and it does not seem that they have tested any alternative. Moreover, our method can use an arbitrary algorithm to solve the problem from Line 4, e.g., Adam or its variations. In addition to being agnostic to the used sub-solver, our approach also supports stateless clients by design, whereas the stateless version of FedSim and FedAlt were tested empirically by Pillutla et al. (2022b) as heuristic modifications.

## 4 Theory

We present the convergence theory for Algorithm 1 and its variants in several cases. First of all, let us understand how its idealistic version, which computes $w_m^*(\theta)$ exactly, would converge under our assumptions.

**Theorem 1.** *Let Assumptions 1-2 hold and assume that we use exact updates, $w_m^r = \arg\min_w f_m(\theta^r, w)$. If we choose $\gamma_\theta \leq \frac{1}{L}$, then*

$$\min_{r < R} \mathbb{E}\left[ \|F(\theta^r)\|^2 \right] \leq \frac{L\|\theta^0 - \theta^*\|^2}{\gamma_\theta R},$$

*where $\theta^*$ is any vector such that $F(\theta^*) = 0$.*

The theorem establishes convergence of operator norms to 0, which implies that the global parameters $\theta^r$ eventually approach $\theta^*$. If we compare the result to the standard bounds on convergence of gradient descent, we can see that there is no slowdown. In this sense, personalization completely removes the issues of data heterogeneity, which usually causes FL method to converge slower (Khaled et al., 2019). In more practical scenarios, Algorithm 1 will compute $w_m^*(\theta)$ inexactly, which will affect the convergence. The next theorem outlines what happens in that case.

**Theorem 2** (Informal). *Let Assumptions 1-2 hold and $f_m$ be smooth and strongly convex in $w$. If $\tau$ is large enough, then*

$$\min_{r < R} \mathbb{E}\left[ \|F(\theta^r)\|^2 \right] \leq \frac{4L\|\theta^0 - \theta^*\|^2}{\gamma_\theta R}.$$

---

**Algorithm 3** Asynchronous FFGG

1: **Input:** initialization $\theta^0 \in \mathbb{R}^d$, stepsizes $\gamma_w, \gamma_\theta > 0$, number of local steps $\tau \in \mathbb{N}$
2: **for** $r = 0, 1, 2, \dots$ **do**
3:     Receive update from client $j_r$ with delay $d_j^r$
4:     $\theta^{r+1} = \theta^r - \gamma_\theta \Delta_{j_r}^{r-d_j^r}$
5:     Sample new client $m_r$ and initialize $w_{m_r}^0$ randomly
6:     Solve $w_{m_r}^r \approx \mathrm{argmin}_w f_{m_r}(\theta^{r+1}, w)$
          E.g., using Algorithm 2
7:     $\Delta_{m_r}^{r+1} = \nabla_1 f_{m_r}(\theta^{r+1}, w_{m_r}^\tau)$
8: **end for**

---

**Algorithm 4** Local FFGG

**Input:** initialization $\theta^0 \in \mathbb{R}^d$, stepsize $\gamma_w, \gamma_\theta > 0$
1: **for** $r = 0, 1, 2, \dots$ **do**
2:     Sample a batch of clients $C^r$
3:     **for** client $m \in C^r$ **do**
4:         $\theta^{r,0} = \theta^r$ and $w_m^{r,0} \approx \mathrm{argmin}_w f_m(\theta^r, w)$
5:         **for** $k = 0, 1, \dots, K-1$ **do**
6:             $w_m^{r,k+1} = w_m^{r,k} - \gamma_w \nabla_2 f_m(\theta^{r,k}, w_m^{r,k})$
7:             $\theta^{r,k+1} = \theta^{r,k} - \gamma_\theta \nabla_1 f_m(\theta^{r,k}, w_m^{r,k+1})$
8:         **end for**
9:         $\Delta_m^r = \theta^r - \theta^{r,K}$
10:     **end for**
11:     $\theta^{r+1} = \theta^r - \frac{1}{|C^r|} \sum_{m \in C^r} \Delta_m^r$
12: **end for**

---

We formulate Theorem 2 more rigorously in the supplementary material; see Theorem 6. The main message, however, remains the same. Moreover, we can see that the local updates *are important*, which is in stark contrast to the theory in (Pillutla et al., 2022b), where having more local updates does not improve convergence.

## 4.1 Breaking the curse in other settings

**Asynchronous communications.** We consider Algorithm 3, asynchronous variant of Algorithm 1. For this method, whenever one of the clients finishes local computations, the server immediately uses the client's update to perform a step, and then that client begins local work starting from newly updated global parameters. The following result holds in this setting.

**Theorem 3** (Informal)**.** *Let Assumptions 1-2 hold and assume that we use exact updates. Assume the delays and the number of active clients are bounded. Then*

$$\min_{r < R} \mathbb{E}\left[\|F(\theta^r)\|^2\right] \leq \frac{2L\|\hat{\theta}^0 - \theta^*\|^2}{\gamma_\theta R},$$

*where $\hat{\theta}^0$ is defined in* (15)*.*

We highlight the fact that the proposed problem formulation allows for Algorithm 1 to converge in an arbitrarily heterogeneous regime. The detailed formulation and proof of this result can be found in the supplementary material, see Theorem 7.

**Byzantine-robust learning.** To achieve Byzantine robustness, our method needs a small but very important modification: in line 7, instead of averaging, we apply *agnostic $(\delta, c)$-robust aggregation* (see Definition 2 in the appendix): $\theta^{r+1} = \theta^r - \gamma_\theta \mathtt{ARAgg}(\Delta_1^r, \Delta_2^r, \dots, \Delta_M^r)$, where $\{1, \dots, M\}$ is the set of all clients, $B \leq \delta M$ of them are Byzantines and $\delta < 1/2$ (see the detailed setup in Appendix D). For this version of the method, we derive the following result.

**Theorem 4** (Informal)**.** *Let operators $F_m$ be $\ell_m$-cocoercive, $F$ be strongly monotone, and good clients compute stochastic estimates satisfying $\mathbb{E}[\|g_m(\theta)\|^2] \leq \rho_{in}\|F_m(\theta)\|^2$. If $\delta$ is small enough, then for Byzantine-robust version of Algorithm 1 we have*

$$\mathbb{E}\left[\|\theta^R - \theta^*\|^2\right] \leq \left(1 - \frac{\gamma_\theta \mu}{2}\right)^R \|\theta^0 - \theta^*\|^2.$$

We emphasize that for arbitrary heterogeneous problems, it is impossible to tolerate even one Byzantine client. In contrast, Theorem 4 implies that for the considered problem formulation, the proposed algorithm converges linearly to the exact solution (asymptotically) even when data on clients is heterogeneous and a small amount of Byzantines takes part in the training. The complete formulation of the above result is deferred to the supplementary material, see Theorem 8.

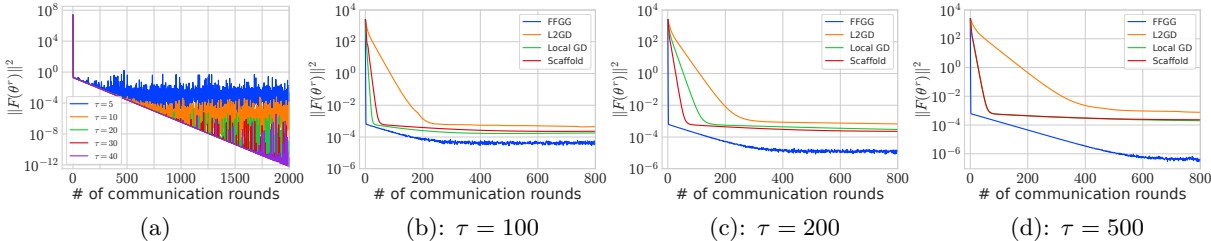

Figure 1: (a): convergence of FFGG varying the number of local steps of conjugate gradients solver $\tau$; (b-d): comparison of FFGG against Scaffold, Local GD, and L2GD varying the number of local gradient steps in each method (denoted as $\tau$ for all methods). Note that unlike in (a), FFGG uses gradient descent as a subsolver to make the comparison fair.

## 4.2 Benefit of cooperation

As discussed in the introduction, we could have chosen to fit the local data on each client individually. However, the drawback of such an approach is that the clients will not benefit from cooperative training. Here, we show that having partial personalization allows to find parameters $\theta$ that are useful for all clients. This implies that in expectation, a new client $m$ that receives $\theta^*$ as initialization will have a better value of post-training risk $f_m(\theta^*, w_m^*(\theta))$. We have the following result.

**Theorem 5.** *Let $F_m$ be monotone for all $m$ and Assumptions 1-2 hold, and assume we found $\theta^*$ such that $\mathbb{E}_m[F_m(\theta^*)] = 0$. Then, $\theta^*$ minimizes the expected risk $\mathbb{E}_m[f_m(\theta, w_m^*(\theta))]$.*

The key observation of Theorem 5 is that finding the solution to $\mathbb{E}[F_m(\theta)] = 0$ is equivalent to minimizing $\mathbb{E}[f_m(\theta, w_m^*(\theta))]$. The obtained $\theta^*$ turns out to be optimal for solving a more challenging bilevel optimization problem.

When we combine the result of Theorem 5 with Assumption 2, we get

$$(\theta^*, w_m^*(\theta)) = \operatorname{argmin}_{\theta, w} f_m(\theta, w) \qquad \text{for all } m.$$

This means that the combined partially personalized model $\theta^*, w_m^*(\theta^*)$ fits the data of client $m$ perfectly. When the model is deployed using the found $\theta^*$, all new clients benefit from having a part of the model pretrained optimally for them. This reduces the computation load, especially if $\theta$ is a big part of the model.

## 4.3 How to split the parameters

The derived theory gives us the following insights:

1. To break the curse of data heterogeneity, the personalized part of the model $w$ must be sufficiently large. Then, Assumption 2 is satisfied, and the training will be easier.

2. If the previous condition is satisfied, Theorem 5 suggests that every client will be able to fit the data perfectly by fine-tuning only $w$. Thus, the smaller the personalized part $w$ is, the easier will be its fine-tuning.

Therefore, the balance between $\theta$ and $w$ when splitting the parameters is crucial. In some cases, it is known that fine-tuning just the batch normalization layers can be sufficient (Li et al., 2020b). Our theory suggests that the key quantity to measure the quality of the split is $\|F_m(\theta)\|$, which we can compute in practice. If it is too close to 0, $\theta$ will not be updated, so one should consider decreasing the number of personalized parameters. It seems less obvious how to detect that we should personalize *more* parameters. Several layers for personalization have been developed in practical works, see Section 5 for an empirical study.

## 5 Experiments

The detailed description of all experimental setups is deferred to the Appendix E.

Table 2: Test accuracy across different model variants and datasets. For FedAlt and FedSim, we report the numbers for the best performing option in each experiment.

| Variant/Test acc. (%) | FEMNIST | GLDv2 | StackOverflow |
|---|---|---|---|
| FedAvg | 93.18 | 51.43 | 23.82 |
| Local Training | $67.29_{\pm 0.15}$ | $18.73_{\pm 0.28}$ | $10.19_{\pm 1.12}$ |
| FFGG (Input Layer) | $93.60_{\pm 0.02}$ | $51.25_{\pm 0.03}$ | $24.11_{\pm 0.02}$ |
| FFGG (Output Layer) | $93.58_{\pm 0.04}$ | $55.20_{\pm 0.04}$ | $\mathbf{24.92}_{\pm 0.01}$ |
| FFGG (Adapter) | $\mathbf{94.26}_{\pm 0.03}$ | $\mathbf{64.93}_{\pm 0.04}$ | $24.80_{\pm 0.01}$ |
| FedAlt (Stateless, Best) | $93.97_{\pm 0.03}$ | $64.10_{\pm 0.14}$ | $\mathbf{24.94}_{\pm 0.01}$ |
| FedSim (Stateless, Best) | $93.89_{\pm 0.02}$ | $63.19_{\pm 0.04}$ | $\mathbf{24.94}_{\pm 0.01}$ |

## 5.1 The more local work, the better the convergence

In our first experiment, we study the convergence of Algorithm 1 with inexact gradient computation. We test it on the problem from Example 1, namely:

$$f_m(\theta, w) = \psi_m(\theta) + \frac{1}{2}\|\mathbf{A}_m\theta + \mathbf{B}_m w - y_m\|^2,$$

$$\psi_m(\theta) = \frac{1}{2}\|\mathbf{H}_m\theta - b_m\|^2,$$

where $\mathbf{H}_m, \mathbf{A}_m \in \mathbb{R}^{n \times d_\theta}, \mathbf{B}_m \in \mathbb{R}^{n \times d_w}, b_m, y_m \in \mathbb{R}^n$ with $n = 10000, d_\theta = 100, d_w = 50$. The number of clients is 32. All matrices are generated from uniform distribution on $[0, 1]$, and then divided by the second dimension (i.e., $d_\theta$ for $\mathbf{H}_m, \mathbf{A}_m$ and $d_w$ for $\mathbf{B}_m$).

We use SciPy's (Virtanen et al., 2020) implementation of the Conjugate Gradient (CG) method to solve local subproblem in $w$ and vary the number of inner steps $\tau$ of CG (see Figure 1, (a)). The convergence with a small number of local steps $\tau = 10$ is already sufficient to achieve an error as small as $10^{-4}$. Moreover, Algorithm 1 converges to the exact solution for $\tau \in \{30, 40\}$, i.e., without finding the precise solution of the local subproblem. This experiment shows that the convergence of FFGG indeed improves with an increasing amount of local work, and the method is overall practical.

## 5.2 Comparison against other methods

Next, we compare FFGG combined with Algorithm 2 as a fine-tuner against non-personalized methods such as Scaffold (Karimireddy et al., 2020b) and Local GD, and fully personalized method L2GD (Hanzely & Richtárik, 2020). For Scaffold, we set outer and inner stepsizes to be equal to 0.5 and $\frac{1}{L_f \tau}$ correspondingly, where $L_f$ is a smoothness constant of $f_m$. For Local GD the stepsize is equal to $\frac{1}{L_f \tau}$. Finally, for L2GD we choose $\lambda = 0.1$ and stepsize to be equal to $(2M)^{-1} \max\left\{L(1-p)^{-1}, \lambda p^{-1}\right\}$, where $p = \tau^{-1}$ (we make such choice for $p$ to make the number of local steps to be close to $\tau$ in expectation).

We test the convergence of the methods changing the number of local steps $\tau \in \{100, 200, 500\}$. Figure 1 (b-d) shows that FFGG outperforms other baselines in all cases. We also highlight that FFGG's convergence improves when we increase the number of local steps as it is predicted by theory.

## 5.3 Comparison on real-world datasets

Finally, we evaluate our method on real-world federated datasets: FEMNIST (character recognition), GLDv2 (Visual Landmark Recognition), and StackOverflow (next word prediction). We demonstrate that FFGG leads to a significant performance improvement compared to non-personalized FedAvg and exhibits better adaptability to heterogeneity compared to methods with the same personalized objective, such as FedAlt (Pillutla et al., 2022b).

As previously discussed, we do not address the optimal parameter split in this work. To achieve partial personalization, we follow the setup from (Pillutla et al., 2022b). However, we show that when *overpersonal-*

*ization* occurs, i.e., we can achieve zero training loss, FFGG outperforms all other methods by a substantial margin. All experimental details and hyperparameter selections are provided in the appendix. For partial personalization, we consider three partitioning schemes:

- *Input-layer personalization*: This architectural design customizes the input layer to learn personalized representations, whereas the remaining part of the model is common to all clients. For predicting the next word, the initial transformer layer is personalized instead of the embedding layer.

- *Output-layer personalization*: This design learns a common representation but customizes the prediction layer. In a transformer model, we adapt the final transformer layer instead of the output layer for personalization.

- *Adapter personalization*: Every client uses a personalized low-rank adapter to fine-tune the global model.

We also introduce an algorithmic extension to Algorithm 1 to incorporate local steps with respect to global parameters into the training loop. After receiving a global model from the server, clients randomly initialize personalized parameters and perform one local epoch with respect to these parameters to approximate $w_m^*(\theta)$. Following this step, we alternate between stochastic gradient steps with respect to global and local parameters. We only initialize $w_m$ at the beginning of local training. This approach allows us to approximate $w_m^*(\theta)$ with a single gradient step after initial fine-tuning. The pseudocode for this algorithm is provided in Algorithm 4. Our results are presented in Table 2. The displayed values represent averages over three independent seeds. It is worth noting that for both datasets, FFGG leads to an improvement of at least one percent in final test accuracy over non-personalized FedAvg. We also observe that Adapter is a particularly useful technique for partially personalizing local models. The largest improvement, exceeding 13%, is observed for FFFG (Adapter) on the GLDv2 dataset. In this particular case, the final train accuracy for all clients is 100%, which aligns well with our theory as it implies that $F_m(\theta^*) = 0$. Additionally, since we employ the same experimental setup for personalization as Pillutla et al. (2022b), we also compare our results with their stateless version of FedAlt (notably, the stateless version is not analyzed in their work). We report the performance of the personalization technique that yielded the best results for FedAlt. It is noteworthy that FFGG consistently performs as well as, if not better than, FedAlt, outperforming FedAlt in two out of three datasets.

## 6 Limitations

In this section, we discuss some limitations of the proposed algorithms and analysis.

- Since the proposed algorithms require the clients to approximately minimize their objective functions w.r.t. their personalized parameters (Line 4, Algorithm 1), it creates a computation and latency overhead in comparison to standard Parallel (S)GD. However, if we use Local-GD (Algorithm 2) as a subsolver for the mentioned problem, then FFGG has no overheads in comparison to standard FedAvg. Moreover, FFGG has even cheaper iterations than FedAvg since FFGG with Local-GD subsolver computes multiple gradients with respect to personalized parameters and one gradient with respect to global parameters, while FedAvg requires the clients to compute multiple gradients with respect to all parameters.

- The proposed methods require knowing the split of parameters, which might be non-trivial to obtain.

- The analysis relies on Assumption 1, which is closely related to convexity and smoothness. These assumptions are commonly used in the literature but are not common in practice, so the main utility of the analysis is to get a general understanding of the methods' behavior.

## 7 Conclusion

We proposed a new analysis of partial personalization that shows its provable benefits in Federated Learning. The main takeaways can be summarized as follows.

- We proved that the problem can always be made *overpersonalized* and the data heterogeneity slow-down can be completely eradicated.

- We also illustrated this by showing that, in contrast to standard FL, asynchronous training with partial personalization converges precisely, and partial personalization can be made Byzantine-robust.

- Our theory also suggests algorithmic changes to how the training should be performed and allows for generic local solvers. Compared to the work of Pillutla et al. (2022b), our methods are stateless, and our theory does not require making stepsizes smaller than $\mathcal{O}\left(1/\tau\right)$, where $\tau$ is the number of local steps. Finally, our assumptions are satisfied for several natural classes of functions, highlighting that our theory is quite general.

There are several open questions that can be of interest to make personalization more practical. First of all, a direction that seems important to us is how we can find optimal splits between $\theta$ and $w$ to achieve both the speed-up of removed data heterogeneity and make sure that clients benefit from cooperation. Secondly, parameter-efficient fine-tuning might bring even more speed-ups. Lastly, while the statistical effect of cooperation was left out of consideration in our work, it can bring new insights and deserves some attention.

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

## A    Deferred proofs

### A.1    Proof of cocoercivity for Example 1

Below we show that the function $f_m(\theta, w) = \phi_m(\theta) + \frac{1}{2}\|\mathbf{A}_m\theta + \mathbf{B}_m w - y_m\|^2$ satisfies Assumption 1.

*Proof.* Notice that $\phi_m(\theta)$ does not depend on $w$, so

$$\nabla_1 f_m(\theta, w_m^*(\theta)) = \nabla\phi_m(\theta) + \mathbf{A}_m^\top(\mathbf{A}_m\theta + \mathbf{B}_m w_m^*(\theta) - y_m).$$

Then, by convexity and $L_\phi$-smoothness of $\phi_m$, we have

$$\langle \nabla_1 f_m(\theta_1, w_m^*(\theta_1)) - \nabla_1 f_m(\theta_2, w_m^*(\theta_2)), \theta_1 - \theta_2\rangle$$
$$= \langle \nabla\phi(\theta_1) - \nabla\phi(\theta_2), \theta_1 - \theta_2\rangle + \langle \mathbf{A}_m^\top(\mathbf{A}_m\theta_1 + \mathbf{B}_m w_m^*(\theta_1) - \mathbf{A}_m\theta_2 - \mathbf{B}_m w_m^*(\theta_2)), \theta_1 - \theta_2\rangle$$
$$\geq \frac{1}{L_\phi}\|\nabla\phi(\theta_1) - \nabla\phi(\theta_2)\|^2 + \langle \mathbf{A}_m^\top(\mathbf{A}_m\theta_1 + \mathbf{B}_m w_m^*(\theta_1) - \mathbf{A}_m\theta_2 - \mathbf{B}_m w_m^*(\theta_2)), \theta_1 - \theta_2\rangle.$$

Let us find the value of $w_m^*(\theta)$. Differentiating $f_m(\theta, w)$ with respect to $w$ and setting the gradient to 0, we get

$$\mathbf{B}_m^\top(\mathbf{A}_m\theta + \mathbf{B}_m w_m^*(\theta) - y_m) = 0,$$

whence

$$\mathbf{B}_m^\top \mathbf{B}_m w_m^*(\theta) = \mathbf{B}_m^\top(y_m - \mathbf{A}_m\theta) \qquad \text{and} \qquad w_m^*(\theta) = \mathbf{B}_m^\dagger(y_m - \mathbf{A}_m\theta).$$

Substituting this into the previous lower bound, we get

$$\langle \nabla_1 f_m(\theta_1, w_m^*(\theta_1)) - \nabla_1 f_m(\theta_2, w_m^*(\theta_2)), \theta_1 - \theta_2\rangle$$
$$\geq \langle \mathbf{A}_m^\top(\mathbf{A}_m\theta_1 - \mathbf{B}_m\mathbf{B}_m^\dagger\mathbf{A}_m\theta_1 - \mathbf{A}_m\theta_2 + \mathbf{B}_m\mathbf{B}_m^\dagger\mathbf{A}_m\theta_2), \theta_1 - \theta_2\rangle + \frac{1}{L_\phi}\|\nabla\phi(\theta_1) - \nabla\phi(\theta_2)\|^2$$
$$= \langle \mathbf{A}_m^\top(\mathbf{I} - \mathbf{B}_m\mathbf{B}_m^\dagger)\mathbf{A}_m(\theta_1 - \theta_2), \theta_1 - \theta_2\rangle + \frac{1}{L_\phi}\|\nabla\phi(\theta_1) - \nabla\phi(\theta_2)\|^2.$$

Let $\mathbf{B}_m = \mathbf{U}\boldsymbol{\Sigma}\mathbf{V}^\top$ be the SVD decomposition of $\mathbf{B}_m$, then $\mathbf{I} - \mathbf{B}_m\mathbf{B}_m^\dagger = \mathbf{I} - \mathbf{U}\boldsymbol{\Sigma}\boldsymbol{\Sigma}^\dagger\mathbf{U}^\top = \mathbf{U}(\mathbf{I} - \boldsymbol{\Sigma}\boldsymbol{\Sigma}^\dagger)\mathbf{U}^\top$ is a symmetric positive semi-definite matrix. Therefore, $\mathbf{A}_m^\top(\mathbf{I} - \mathbf{B}_m\mathbf{B}_m^\dagger)\mathbf{A}_m$ is symmetric positive semi-definite as well. Thus, we obtain that the linear term in $F_m$ is convex. Since it is the gradient of a quadratic, it is $\hat{L}$-smooth with $\hat{L} = \|\mathbf{A}_m^\top(\mathbf{I} - \mathbf{B}_m\mathbf{B}_m^\dagger)\mathbf{A}_m\|$. Therefore,

$$\langle \nabla_1 f_m(\theta_1, w_m(\theta_1)) - \nabla_1 f_m(\theta_2, w_m(\theta_2)), \theta_1 - \theta_2\rangle$$
$$\geq \frac{1}{L_\phi}\|\nabla\phi(\theta_1) - \nabla\phi(\theta_2)\|^2 + \frac{1}{\hat{L}}\|\mathbf{A}_m^\top(\mathbf{I} - \mathbf{B}_m\mathbf{B}_m^\dagger)\mathbf{A}_m(\theta_1 - \theta_2)\|^2$$
$$\geq \frac{1}{2\max(L_\phi, \hat{L})}\|\nabla\phi(\theta_1) - \nabla\phi(\theta_2) + \mathbf{A}_m^\top(\mathbf{I} - \mathbf{B}_m\mathbf{B}_m^\dagger)\mathbf{A}_m(\theta_1 - \theta_2)\|^2$$
$$= \frac{1}{2\max(L_\phi, \hat{L})}\|\nabla_1 f_m(\theta_1, w_m(\theta_1)) - \nabla_1 f_m(\theta_2, w_m(\theta_2))\|^2.$$

which is exactly $\frac{1}{2\max(L_\phi, \hat{L})}$-cocoercivity of $F_m$.    □

### A.2    Proof of cocoercivity for Example 2

Now, we study a general function $f_m$ that has bounded derivatives.

*Proof.* Let us lower bound the Jacobian of $F_m$:

$$\nabla_\theta F_m(\theta) = \nabla_{11}^2 f_m(\theta, w_m^*(\theta)) + \nabla_\theta w_m^*(\theta)\nabla_{12}^2 f_m(\theta, w_m^*(\theta))$$
$$\succcurlyeq \mu\mathbf{I} + \nabla_\theta w_m^*(\theta)\nabla_{12}^2 f_m(\theta, w_m^*(\theta))$$
$$\succcurlyeq \mu\mathbf{I} - \|\nabla_\theta w_m^*(\theta)\| \cdot \|\nabla_{12}^2 f_m(\theta, w_m^*(\theta))\|$$
$$\succcurlyeq \frac{\mu}{2}\mathbf{I}.$$

This lower bound implies $\frac{\mu}{2}$-strong monotonicity. We also have a similar upper bound:

$$\nabla_\theta F_m(\theta) = \nabla_{11}^2 f_m(\theta, w_m^*(\theta)) + \nabla_\theta w_m^*(\theta)\nabla_{12}^2 f_m(\theta, w_m^*(\theta)) \preccurlyeq (L + C_1 C_2)\mathbf{I}.$$

Since $L \geq \mu$, we have $C_1 C_2 \leq \frac{\mu}{2} \leq L$, and

$$\nabla_\theta F_m(\theta) \preccurlyeq 2L\mathbf{I},$$

which implies $F_m$ is $(2L)$-Lipschitz. Combining this with strong monotonicity, we get that it is also $\frac{\mu}{4L^2}$-cocoercive. □

### A.3 Proof of cocoercivity for Example 3

Example 3 is the hardest to study. First, we state and prove the following standard result.

**Proposition 2.** Let $\varphi_1, \varphi_2$ be $L$-smooth convex functions. Then $\varphi = \varphi_1 + \varphi_2$ is $(2L)$-cocoercive.

*Proof.* As can be found in standard textbooks, such as (Nesterov, 2013), convexity and smoothness imply that both $\varphi_1$ and $\varphi_2$ are $L$-cocoercive. Moreover, for any $\theta_1, \theta_2$, it holds

$$\begin{aligned}
\langle \nabla\varphi(\theta_1) - \nabla\varphi(\theta_2), \theta_1 - \theta_2 \rangle &= \langle \nabla\varphi_1(\theta_1) - \nabla\varphi_1(\theta_2), \theta_1 - \theta_2 \rangle + \langle \nabla\varphi_2(\theta_1) - \nabla\varphi_2(\theta_2), \theta_1 - \theta_2 \rangle \\
&\geq \frac{1}{L}\|\nabla\varphi_1(\theta_1) - \nabla\varphi_1(\theta_2)\|^2 + \frac{1}{L}\|\nabla\varphi_2(\theta_1) - \nabla\varphi_2(\theta_2)\|^2 \\
&\geq \frac{1}{2L}\|\nabla\varphi_1(\theta_1) - \nabla\varphi_1(\theta_2) + \nabla\varphi_2(\theta_1) - \nabla\varphi_2(\theta_2)\|^2 \\
&= \frac{1}{2L}\|\nabla\varphi(\theta_1) - \nabla\varphi(\theta_2)\|^2,
\end{aligned}$$

which is exactly what we need to prove. □

Now we proceed to prove cocoercivity of $F_m$ from Example 3.

*Proof.* Notice that $\phi_m(\theta)$ does not depend on $w$, so

$$\nabla_1 f_m(\theta, w_m^*(\theta)) = \nabla\phi_m(\theta) + \mathbf{A}_m^\top \nabla\psi_m(\mathbf{A}_m\theta + \mathbf{B}_m w_m^*(\theta) - y_m).$$

Let us find the value of $w_m^*(\theta)$. Differentiating $f_m(\theta, w)$ with respect to $w$ and setting the gradient to 0, we get

$$\mathbf{B}_m^\top \nabla\psi_m(\mathbf{A}_m\theta + \mathbf{B}_m w_m^*(\theta) - y_m) = 0.$$

Let $u_1 = \mathbf{A}_m\theta_1 + \mathbf{B}_m w_m^*(\theta_1) - y_m$ and $u_2 = \mathbf{A}_m\theta_2 + \mathbf{B}_m w_m^*(\theta_2) - y_m$. Then,

$$\begin{aligned}
&\langle \mathbf{A}_m^\top \nabla\psi_m(\mathbf{A}_m\theta_1 + \mathbf{B}_m w_m^*(\theta_1) - y_m) - \mathbf{A}_m^\top \nabla\psi_m(\mathbf{A}_m\theta_2 + \mathbf{B}_m w_m^*(\theta_2) - y_m), \theta_1 - \theta_2 \rangle \\
&= \langle \mathbf{A}_m^\top \nabla\psi_m(u_1) - \mathbf{A}_m^\top \nabla\psi_m(u_2), \theta_1 - \theta_2 \rangle \\
&= \langle \nabla\psi_m(u_1) - \nabla\psi_m(u_2), \mathbf{A}_m(\theta_1 - \theta_2) \rangle \\
&= \langle \nabla\psi_m(u_1) - \nabla\psi_m(u_2), (\mathbf{A}_m\theta_1 - y_m) - (\mathbf{A}_m\theta_2 - y_m) \rangle \\
&= \langle \nabla\psi_m(u_1) - \nabla\psi_m(u_2), (\mathbf{A}_m\theta_1 + \mathbf{B}_m w_m^*(\theta_1) - y_m) - (\mathbf{A}_m\theta_2 + \mathbf{B}_m w_m^*(\theta_2) - y_m) \rangle \\
&\quad - \langle \nabla\psi_m(u_1) - \nabla\psi_m(u_2), \mathbf{B}_m w_m^*(\theta_1) - \mathbf{B}_m w_m^*(\theta_2) \rangle.
\end{aligned}$$

Moreover, since $\mathbf{B}_m^\top \nabla\psi_m(u_1) = 0$ and $\mathbf{B}_m^\top \nabla\psi_m(u_2) = 0$, we have

$$\begin{aligned}
&\langle \nabla\psi_m(u_1) - \nabla\psi_m(u_2), \mathbf{B}_m w_m^*(\theta_1) - \mathbf{B}_m w_m^*(\theta_2) \rangle \\
&= \langle \mathbf{B}_m^\top \nabla\psi_m(u_1) - \mathbf{B}_m^\top \nabla\psi_m(u_2), w_m^*(\theta_1) - w_m^*(\theta_2) \rangle \\
&= 0.
\end{aligned}$$

Plugging this back, we get

$$\langle \mathbf{A}_m^\top \nabla \psi_m(u_1) - \mathbf{A}_m^\top \nabla \psi_m(u_2), \theta_1 - \theta_2 \rangle = \langle \nabla \psi_m(u_1) - \nabla \psi_m(u_2), u_1 - u_2 \rangle$$
$$\geq \frac{1}{L_\psi} \| \nabla \psi_m(u_1) - \nabla \psi_m(u_2) \|^2.$$

Therefore, $\nabla_1 f_m(\theta, w_m^*(\theta))$ is equal to the sum of two cocoercive operators. Thus, $F_m$ is cocoercive as well. $\qquad\square$

### A.4 Proof of Theorem 1

*Proof.* Since we assume exact computation of $w^*(\theta^r)$ for all $r$ and $m \in C^r$, it holds

$$\Delta_m^r = \nabla_1 f_m(\theta^r, w_m^*(\theta^r)) = F_m(\theta^r).$$

Therefore, we have the following recursion:

$$\| \theta^{r+1} - \theta^* \|^2 = \| \theta^r - \theta^* \|^2 - \frac{2\gamma_\theta}{|C^r|} \sum_{m \in C^r} \langle F_m(\theta^r), \theta^r - \theta^* \rangle + \left\| \frac{\gamma_\theta}{|C^r|} \sum_{m \in C^r} F_m(\theta^r) \right\|^2$$
$$\overset{(4)}{\leq} \| \theta^r - \theta^* \|^2 - \frac{2\gamma_\theta}{L|C^r|} \sum_{m \in C^r} \| F_m(\theta^r) \|^2 + \left\| \frac{\gamma_\theta}{|C^r|} \sum_{m \in C^r} F_m(\theta^r) \right\|^2$$
$$\leq \| \theta^r - \theta^* \|^2 - \frac{2\gamma_\theta}{L|C^r|} \sum_{m \in C^r} \| F_m(\theta^r) \|^2 + \frac{\gamma_\theta^2}{|C^r|} \sum_{m \in C^r} \| F_m(\theta^r) \|^2$$
$$\overset{\gamma_\theta \leq \frac{1}{L}}{\leq} \| \theta^r - \theta^* \|^2 - \frac{\gamma_\theta}{L|C^r|} \sum_{m \in C^r} \| F_m(\theta^r) \|^2.$$

Taking expectation, we get

$$\mathbb{E}\left[ \| F(\theta^r) \|^2 \right] \leq \mathbb{E}\left[ \frac{1}{|C^r|} \sum_{m \in C^r} \| F_m(\theta^r) \|^2 \right] \leq \frac{L}{\gamma_\theta} \left( \| \theta^r - \theta^* \|^2 - \| \theta^{r+1} - \theta^* \|^2 \right).$$

Summing this bound over $r = 0, \ldots, R-1$, we get

$$\min_{r < R} \mathbb{E}\left[ \| F(\theta^r) \|^2 \right] \leq \frac{1}{R} \sum_{r=0}^{R-1} \mathbb{E}\left[ \| F(\theta^r) \|^2 \right] \leq \frac{L \| \theta^0 - \theta^* \|^2 - \| \theta^R - \theta^* \|^2}{\gamma_\theta R} \leq \frac{L \| \theta^0 - \theta^* \|^2}{\gamma_\theta R},$$

which completes the proof. $\qquad\square$

### A.5 Proof of Theorem 5

*Proof.* Our goal is to show that all solutions to $F(\theta) = 0$ are also minimizers of $\mathbb{E}[f_m(\theta, w_m^*(\theta))]$. Since we assume that the functions are convex, it is sufficient to show that the gradient of $\mathbb{E}[f_m(\theta, w_m^*(\theta))]$ is equal to 0. We have

$$\nabla_\theta f_m(\theta, w_m^*(\theta)) = \nabla_1 f_m(\theta, w_m^*(\theta)) + \nabla_{12} f(\theta, w_m^*(\theta)) [\nabla_{22}^2 f(\theta, w_m^*(\theta))]^{-1} \nabla_2 f_m(\theta, w_m^*(\theta)).$$

Note that by definition of $w_m^*(\theta)$, it holds $\nabla_2 f_m(\theta, w_m^*(\theta)) = 0$ since $w_m^*(\theta)$ is optimal when the first argument of $f_m$ is fixed. Therefore,

$$\nabla_\theta f_m(\theta, w_m^*(\theta)) = \nabla_1 f_m(\theta, w_m^*(\theta)).$$

Let $\theta^*$ be such that $F(\theta^*) = 0$. Then,

$$F(\theta) = \mathbb{E}[\nabla_1 f_m(\theta, w_m^*(\theta))] = \nabla_\theta \mathbb{E}[f_m(\theta, w_m^*(\theta))] = 0.$$

$\qquad\square$

# B    Algorithm 1 with inexact gradient computation

We consider Algorithm 1 where $\Delta_m^r \neq F_m(\theta^r)$, i.e., with inexact gradient computation. The analysis of the inexact version of Algorithm 1 requires additional assumptions on the problem which are listed below.

**Assumption 3.** *There exist constants $L_w$ and $\mu_w$ such that for any client $m$, the loss $f_m$ is $L_w$-Lipschitz continuous and $\mu_w$-strongly convex in $w$ for any fixed $\theta$, i.e.,*

$$\|\nabla_1 f_m(\theta, w_1) - \nabla_1 f_m(\theta, w_2)\| \leq L_w \|w_1 - w_2\| \tag{5}$$

$$f_m(\theta, w_1) \geq f_m(\theta, w_2) + \langle \nabla_2 f_m(\theta, w_2), w_1 - w_2 \rangle + \frac{\mu_w}{2} \|w_1 - w_2\|^2. \tag{6}$$

If Assumption 3 holds, then the standard result for Gradient Descent in $w$ takes place

$$\|w_m^{r,\tau} - w_m^*(\theta^r)\|^2 \leq (1 - \mu_w \gamma_w)\|w_m^{r,\tau-1} - w_m^*(\theta^r)\|^2 \leq (1 - \mu_w \gamma_w)^\tau \|w_m^{r,0} - w_m^*(\theta^r)\|^2, \tag{7}$$

where the inner stepsize $\gamma_w \leq \frac{1}{L_w}$.

**Assumption 4.** *There exist constants $A, C$ such that for any client $m$, the solution $w_m^*(\theta)$ satisfies*

$$\|w_m^*(\theta)\| \leq A\|\theta - \theta^*\| + C. \tag{8}$$

This assumption holds if the norm of $\nabla_\theta w_m^*(\theta)$ is bounded by $A$, because then $w_m^*$ is $A$-Lipschitz continuous, and consequently $w_m^*$ satisfies Assumption 4:

$$\|w_m^*(\theta)\| \leq \|w_m^*(\theta) - w_m^*(\theta^*)\| + \|w_m^*(\theta^*)\| \leq A\|\theta - \theta^*\| + \|w_m^*(\theta^*)\|.$$

**Remark 1.** For simplicity of explanation, let $w_m^{r,0}$ be initialized as zero and the cardinality of $C^r$ is fixed.

**Remark 2.** We provide the proof of Algorithm 1 where fine-tuning is performed using Local GD (Algorithm 2). In fact, all clients may utilize any other method to solve a subproblem to approximate $w_m^*(\theta^r)$. The only difference in the analysis is that we need to require $\|w_m^{r,\tau} - w_m^*(\theta^r)\|^2 \leq (1-\rho)\|w_m^{r,0} - w_m^*(\theta^r)\|^2$ and assume that $\rho$ is not too small or $\tau$ is sufficiently large in order to derive a convergence. For example, in the case of Example 1, we may use the Conjugate Gradient method, which is more suitable for quadratic problems in $w$.

**Theorem 6.** *Let Assumptions 1, 3 and 4 hold, set the stepsizes as $\gamma_\theta = \frac{1}{L}, \gamma_w = \frac{1}{L_w}$, and assume that the number of local iterations $\tau$ is lower bounded as*

$$\tau \geq \frac{L_w}{\mu_w} \max\left\{2\log aR, 2\log \frac{bR}{r_0}, \log \frac{b^2 R}{r_0^2}\right\},$$

*where $a$ and $b$ are defined as $a := \frac{2L_w A}{L}, \quad b := \frac{2L_w C}{L}$. Then*

$$\min_{r < R} \mathbb{E}\left[\|F(\theta^r)\|^2\right] \leq \frac{4L\|\theta^0 - \theta^*\|^2}{\gamma_\theta R}.$$

*Proof.* Due to inexactness of the update, $\Delta_m^r = \nabla_1 f_m(\theta^r, w_m^{r,\tau}) \neq F_m(\theta^r)$. Thus, $\theta$ will be updated with a biased estimate of $F_m(\theta^r)$. We start with unrolling $\|\theta^{r+1} - \theta^*\|^2$:

$$
\begin{aligned}
\|\theta^{r+1} - \theta^*\|^2 &= \|\theta^r - \theta^*\|^2 - \frac{2\gamma_\theta}{|C^r|} \sum_{m \in C^r} \langle \Delta_m^r, \theta^r - \theta^* \rangle + \left\| \frac{\gamma_\theta}{|C^r|} \sum_{m \in C^r} \Delta_m^r \right\|^2 \\
&= \|\theta^r - \theta^*\|^2 - \frac{2\gamma_\theta}{|C^r|} \sum_{m \in C^r} \langle F_m(\theta^r), \theta^r - \theta^* \rangle - \frac{2\gamma_\theta}{|C^r|} \sum_{m \in C^r} \langle \Delta_m^r - F_m(\theta^r), \theta^r - \theta^* \rangle \\
&\quad + \left\| \frac{\gamma_\theta}{|C^r|} \sum_{m \in C^r} [\Delta_m^r - F_m(\theta^r) + F_m(\theta^r)] \right\|^2 \\
&\overset{(4)}{\leq} \|\theta^r - \theta^*\|^2 - \frac{2\gamma_\theta}{|C^r|} \sum_{m \in C^r} \|F_m(\theta^r)\|^2 - \frac{2\gamma_\theta}{|C^r|} \sum_{m \in C^r} \langle \Delta_m^r - F_m(\theta^r), \theta^r - \theta^* \rangle \\
&\quad + \frac{2\gamma_\theta^2}{|C^r|} \sum_{m \in C^r} \|\Delta_m^r - F_m(\theta^r)\|^2 + \frac{2\gamma_\theta^2}{|C^r|} \sum_{m \in C^r} \|F_m(\theta^r)\|^2 \\
&\overset{\gamma_\theta \leq \frac{1}{2L}}{\leq} \|\theta^r - \theta^*\|^2 - \frac{\gamma_\theta}{L|C^r|} \sum_{m \in C^r} \|F_m(\theta^r)\|^2 - \frac{2\gamma_\theta}{|C^r|} \sum_{m \in C^r} \langle \Delta_m^r - F_m(\theta^r), \theta^r - \theta^* \rangle \\
&\quad + \frac{2\gamma_\theta^2}{|C^r|} \sum_{m \in C^r} \|\Delta_m^r - F_m(\theta^r)\|^2,
\end{aligned}
\tag{9}
$$

where in the first inequality, we also use Young's inequality two times. Now we handle the third term in (9) taking expectation w.r.t to all probability events happened before iteration $r$:

$$
\begin{aligned}
-\frac{2\gamma_\theta}{|C^r|} \sum_{m \in C^r} \langle \Delta_m^r - F_m(\theta^r), \theta^r - \theta^* \rangle &\leq \frac{2\gamma_\theta}{|C^r|} \sum_{m \in C^r} \|\theta^r - \theta^*\| \cdot \|\Delta_m^r - F_m(\theta^r)\| \\
&\leq \frac{2\gamma_\theta}{|C^r|} \sum_{m \in C^r} \|\theta^r - \theta^*\| \sqrt{\|\nabla_1 f_m(\theta^r, w_m^{r,\tau}) - \nabla_1 f_m(\theta^r, w_m^*(\theta^r))\|^2} \\
&\overset{(5)}{\leq} \frac{2\gamma_\theta L_w}{|C^r|} \sum_{m \in C^r} \|\theta^r - \theta^*\| \sqrt{\|w_m^{r,\tau} - w_m^*(\theta^r)\|^2} \\
&\overset{(7)}{\leq} \frac{2\gamma_\theta L_w}{|C^r|} (1 - \gamma_w \mu_w)^{\tau/2} \sum_{m \in C^r} \|\theta^r - \theta^*\| \cdot \|w_m^*(\theta^r)\| \\
&\overset{(4)}{\leq} 2\gamma_\theta L_w (1 - \gamma_w \mu_w)^{\tau/2} (A\|\theta^r - \theta^*\|^2 + C\|\theta^r - \theta^*\|),
\end{aligned}
\tag{10}
$$

where we use the assumption $w_m^{r,0} = 0$. Now we work on the last term in (9)

$$
\begin{aligned}
\frac{2\gamma_\theta^2}{|C^r|} \sum_{m \in C^r} \|\Delta_m^r - F_m(\theta^r)\|^2 &= \frac{2\gamma_\theta^2}{|C^r|} \sum_{m \in C^r} \|\nabla_1 f_m(\theta^r, w_m^{r,\tau}) - \nabla_1 f_m(\theta^r, w_m^*(\theta^r))\|^2 \\
&\overset{(5)}{\leq} \frac{2L_w^2 \gamma_\theta^2}{|C^r|} \sum_{m \in C^r} \|w_m^*(\theta^r) - w_m^{r,\tau}\|^2 \\
&\overset{(7)}{\leq} \frac{2L_w^2 \gamma_\theta^2}{|C^r|} (1 - \gamma_w \mu_w)^\tau \sum_{m \in C^r} \|w_m^*(\theta^r)\|^2 \\
&\overset{(4)}{\leq} 2L_w^2 \gamma_\theta^2 (1 - \gamma_w \mu_w)^\tau (2A^2\|\theta^r - \theta^*\|^2 + 2C^2).
\end{aligned}
\tag{11}
$$

Plugging (10) and (11) in (9), we get

$$\mathbb{E}_r \|\theta^{r+1} - \theta^*\|^2 \le \|\theta^r - \theta^*\|^2 - \frac{\gamma_\theta}{L|C^r|} \sum_{m \in C^r} \mathbb{E}_r \|F_m(\theta^r)\|^2$$
$$+ 2\gamma_\theta L_w (1 - \gamma_w \mu_w)^{\tau/2} (A\|\theta^r - \theta^*\|^2 + C\|\theta^r - \theta^*\|)$$
$$+ 2L_w^2 \gamma_\theta^2 (1 - \gamma_w \mu_w)^\tau (2A^2\|\theta^r - \theta^*\|^2 + C^2).$$

Thus, taking full expectation, we have

$$0 \le \mathbb{E}\left[\frac{\gamma_\theta}{RL|C^r|} \sum_{r=0}^{R-1} \sum_{m \in C^r} \|F_m(\theta^r)\|^2\right] \le \frac{1}{R}(\|\theta^0 - \theta^*\|^2 - \mathbb{E}\|\theta^R - \theta^*\|^2)$$
$$+ \frac{2\gamma_\theta L_w}{R}(1 - \gamma_w \mu_w)^{\tau/2} \sum_{r=0}^{R-1} (A\mathbb{E}\|\theta^r - \theta^*\|^2 + C\sqrt{\mathbb{E}\|\theta^r - \theta^*\|^2})$$
$$+ \frac{2\gamma_\theta^2 L_w^2}{R}(1 - \gamma_w \mu_w)^\tau \sum_{r=0}^{R-1}(2A^2 \mathbb{E}\|\theta^r - \theta^*\|^2 + 2C^2)$$
$$\le \frac{1}{R}(\|\theta^0 - \theta^*\|^2 - \mathbb{E}\|\theta^R - \theta^*\|^2)$$
$$+ \frac{1}{R}\left(2\gamma_\theta L_w A(1 - \gamma_w \mu_w)^{\tau/2} + 4\gamma_\theta^2 L_w^2 A^2 (1 - \gamma_w \mu_w)^\tau\right) \sum_{r=0}^{R-1} \mathbb{E}\|\theta^r - \theta^*\|^2$$
$$+ \frac{2\gamma_\theta L_w C}{R}(1 - \gamma_w \mu_w)^{\tau/2} \sum_{r=0}^{R-1} \sqrt{\mathbb{E}\|\theta^r - \theta^*\|^2} + 4\gamma_\theta^2 L_w^2 C^2 (1 - \gamma_w \mu_w)^\tau. \tag{12}$$

This implies that

$$\mathbb{E}\|\theta^R - \theta^*\|^2 \le \|\theta^0 - \theta^*\|^2$$
$$+ \left(2\gamma_\theta L_w A(1 - \gamma_w \mu_w)^{\tau/2} + 4\gamma_\theta^2 L_w^2 A^2 (1 - \gamma_w \mu_w)^\tau\right) \sum_{r=0}^{R-1} \mathbb{E}\|\theta^r - \theta^*\|^2$$
$$+ 2\gamma_\theta L_w C(1 - \gamma_w \mu_w)^{\tau/2} \sum_{r=0}^{R-1} \sqrt{\mathbb{E}\|\theta^r - \theta\|^2} + 4R\gamma_\theta^2 L_w^2 C^2 (1 - \gamma_w \mu_w)^\tau$$
$$\le \|\theta^0 - \theta^*\|^2 + \left(ae^{-\gamma_w \mu_w \tau/2} + a^2 e^{-\gamma_w \mu_w \tau}\right) \sum_{r=0}^{R-1} \mathbb{E}\|\theta^r - \theta^*\|^2$$
$$+ be^{-\gamma_w \mu_w \tau/2} \sum_{r=0}^{R-1} \sqrt{\mathbb{E}\|\theta^r - \theta^*\|^2} + b^2 e^{-\gamma_w \mu_w \tau} R, \tag{13}$$

where (we plug in stepsize values from the statement and use inequality $(1 - x)^\alpha \le e^{-\alpha x}$)

$$a := \frac{2L_w A}{L}, \quad b := \frac{2L_w C}{L}.$$

Now we will show by induction that $\mathbb{E}\|\theta^r - \theta^*\|^2 \le 4r_0^2$, where $r_0^2 \ge \|\theta^0 - \theta^*\|^2$, for any $r$. The base of induction is trivial since $\|\theta - \theta^*\|^2 \le r_0^2 < 4r_0^2$. Assume that $\mathbb{E}\|\theta^r - \theta^*\|^2 \le r_0^2$ for all $r \in \{0, \ldots, R-1\}$, then it also holds for $\mathbb{E}\|\theta^R - \theta^*\|^2$. Indeed, the restriction on $\tau$

$$\tau \ge \max\left\{\frac{2L_w \log aR}{\mu_w}, \frac{2L_w \log \frac{bR}{r_0}}{\mu_w}, \frac{L_w \log \frac{b^2 R}{r_0^2}}{\mu_w}\right\}$$

implies that by the basis of induction, the following:

$$ae^{-\gamma_w\mu_w\tau/2}\sum_{r=0}^{R-1}\mathbb{E}\|\theta^r-\theta^*\|^2 \le a\exp\left(-\gamma_w\mu_w\tau/2\right)Rr_0^2$$
$$\le a\exp\left(-\frac{\mu_w}{2L_w}\frac{2L_w\log aR}{\mu_w}\right)Rr_0^2$$
$$= a\exp(-\log aR)Rr_0^2 = \frac{aRr_0^2}{aR} = r_0^2.$$

Similarly,

$$a^2e^{-\gamma_w\mu_w\tau}\sum_{r=0}^{R-1}\mathbb{E}\|\theta^r-\theta^*\|^2 \le a^2\exp(-\gamma_w\mu_w\tau)Rr_0^2$$
$$= (a\exp(-\gamma_w\mu_w\tau/2))^2Rr_0^2$$
$$\le \left(a\exp\left(-\frac{\mu_w}{2L_w}\frac{2L_w\log aR}{\mu_w}\right)\right)^2Rr_0^2$$
$$= (a\exp(\log(-aR)))^2Rr_0^2 = \frac{a^2Rr_0^2}{a^2R^2} \le r_0^2.$$

And we have the same bounds for the remaining two terms in (13):

$$be^{-\gamma_w\mu_w\tau/2}\sum_{r=0}^{R-1}\sqrt{\mathbb{E}\|\theta^r-\theta^*\|^2} \le b\exp(-\gamma_w\mu_w\tau/2)Rr_0$$
$$\le b\exp\left(-\frac{\mu_w}{2L_w}\frac{2L_w\log\frac{bR}{r_0}}{\mu_w}\right)Rr_0$$
$$= b\exp\left(-\log\frac{bR}{r_0}\right)Rr_0 = \frac{bRr_0}{bR/r_0} = r_0^2,$$

and

$$b^2e^{-\gamma_w\mu_w\tau}R \le b^2\exp\left(-\frac{\mu_w}{L_w}\frac{L_w\log\frac{b^2R}{r_0^2}}{\mu_w}\right)R$$
$$= b^2\exp\left(-\frac{b^2R}{r_0^2}\right)R = \frac{b^2R}{b^2R/r_0^2} = r_0^2.$$

Hence, all four terms in (13) are smaller than $r_0^2$. Thus, with such a choice of stepsize, we prove the statement of induction. Note that restrictions on $\tau$ logarithmically depend on $R$ only, hence it is not a strong assumption.

Now we establish the statement of the theorem. Using the statement of induction in (12), we get

$$\min_{r<R}\mathbb{E}\left[\|F(\theta^r)\|^2\right] \le \frac{1}{R}\sum_{r=0}^{R-1}\mathbb{E}\left[\|F(\theta^r)\|^2\right]$$
$$\le \frac{1}{R}\sum_{r=0}^{R-1}\frac{1}{|C^r|}\sum_{m\in C^r}\mathbb{E}\left[\|F_m(\theta^r)\|^2\right]$$
$$\le \frac{4L\|\theta^0-\theta^*\|^2}{\gamma_\theta R}.$$

$\square$

## C  Asynchronous method

We formulate and prove the convergence of Algorithm 3 (which is the asynchronous version of Algorithm 1) with exact computations only, i.e., in line 7 the subproblem is solved exactly. However, the convergence of the inexact version can be derived in a similar way as for Algorithm 1 in Appendix B. As defined in Algorithm 3, we denote the client that finishes the computation at iteration $r$ as $j_r$ and the newly sampled client as $m_r$.

To prove the convergence, we define $\mathrm{prev}(m, r) \coloneqq \max\{j < r : m_j = m\}$ — the last iteration before iteration $r$ when the update from client $m$ was applied. Our analysis is based on the *virtual* iterates, also known as *perturbed* iterates, that were introduced by Mania et al. (2017). In particular, we consider the sequence defined recursively as

$$\hat{\theta}^{r+1} = \hat{\theta}^r - \gamma_\theta F_{m_r}(\theta^r). \tag{14}$$

We also use initialization

$$\hat{\theta}^0 = \theta^0 - \sum_{m \in C^0} \gamma_\theta F_m(\theta^0) \tag{15}$$

since all are initially sampled, clients will compute their gradients using $\theta^0$. The set of active clients $C^r$ initialized with $C^0$ is updated according to $C^{r+1} = \{m_r\} \cup (C^r \setminus \{j_r\})$. We assume that $C^r$ is always bounded, which is satisfied, for instance, when the total number of clients is finite. We assume that delays are bounded.

**Assumption 5.** *There exists a constant $\tau_{\max}$ such that for any client $m$ at iteration $r$ the following inequality holds: $|\mathrm{prev}(m, r) - r| \le \tau_{\max}$, i.e., all the delays are bounded by $\tau_{\max}$.*

**Theorem 7.** *Assume Assumptions 1 and 5 hold. Let the number of active clients is upper bounded by $M$. Assume the stepsize is such that $\gamma_\theta \le (2L\sqrt{2M\tau_{\max}})^{-1}$. Then*

$$\min_{r < R} \mathbb{E}\left[\|F(\theta^r)\|^2\right] \le \frac{2L\|\hat{\theta}^0 - \theta^*\|^2}{\gamma_\theta R}.$$

*Proof.* Let us first show the link between $\hat{\theta}^r$ and $\theta^r$ by induction:

$$\theta^r - \hat{\theta}^r = \sum_{m \in C^r} \gamma_\theta F_m(\theta^{\mathrm{prev}(m,r)}). \tag{16}$$

It is true for the base $r = 0$. Let us assume that it holds for $r - 1$ and prove for $r$. We have

$$\begin{aligned}
\theta^r - \hat{\theta}^r &= \left(\theta^{r-1} - \gamma_\theta F_{j_{r-1}}(\theta^{\mathrm{prev}(m,r-1)})\right) - \left(\hat{\theta}^{r-1} - \gamma_\theta F_{m_{r-1}}(\theta^{r-1})\right) \\
&= \sum_{m \in C^{r-1}} \gamma_\theta F_m(\theta^{\mathrm{prev}(m,r-1)}) + \gamma_\theta(F_{m_{r-1}}(\theta^{r-1}) - F_{j_{r-1}}(\theta^{\mathrm{prev}(m,r-1)})).
\end{aligned}$$

We also remind the reader that $\mathrm{prev}(m_{r-1}, r) = r - 1$ and $C^r = \{m_{r-1}\} \cup (C^{r-1} \setminus \{j_{r-1}\})$. Moreover, for the rest of active workers $m$ (those gradients still have not been applied) we have $\mathrm{prev}(m, r - 1) = \mathrm{prev}(m, r)$. Thus, the above can be rewritten as

$$\theta^r - \hat{\theta}^r = \sum_{m \in C^r} \gamma_\theta F_m(\theta^{\mathrm{prev}(m,r)}).$$

Note that $|C^r| \le M$ by the assumption of the theorem. This lemma says that the difference between $\theta^r$ and $\hat{\theta}^r$ is always equal to the sum of gradients that are being computed at iteration $r$. Having this link, we

continue as follows

$$\mathbb{E}_r\left[\|\hat{\theta}^{r+1}-\theta^*\|^2\right] = \|\hat{\theta}^r-\theta^*\|^2 - 2\gamma_\theta\mathbb{E}_r\left[\langle F_{m_r}(\theta^r),\hat{\theta}^r-\theta^*\rangle\right] + \gamma_\theta^2\mathbb{E}_r\left[\|F_{m_r}(\theta^r)\|^2\right]$$
$$= \|\hat{\theta}^r-\theta^*\|^2 - 2\gamma_\theta\mathbb{E}_r\left[\langle F_{m_r}(\theta^r),\theta^r-\theta^*\rangle\right] + \gamma_\theta^2\mathbb{E}_r\left[\|F_{m_r}(\theta^r)\|^2\right]$$
$$\quad + 2\gamma_\theta\langle F(\theta^r),\theta^r-\hat{\theta}^r\rangle$$
$$= \|\hat{\theta}^r-\theta^*\|^2 - \frac{3\gamma_\theta}{2}\mathbb{E}_r\left[\langle F_{m_r}(\theta^r),\theta^r-\theta^*\rangle\right] - \frac{\gamma_\theta}{2}\mathbb{E}_r\left[\langle F_{m_r}(\theta^r),\theta^r-\theta^*\rangle\right]$$
$$\quad + \gamma_\theta^2\mathbb{E}_r\left[\|F_{m_r}(\theta^r)\|^2\right] + 2\gamma_\theta\langle F(\theta^r),\theta^r-\hat{\theta}^r\rangle$$
$$\overset{(i)}{\leq} \|\hat{\theta}^r-\theta^*\|^2 - \frac{3\gamma_\theta}{2}\mathbb{E}_r\left[\langle F_{m_r}(\theta^r),\theta^r-\theta^*\rangle\right] - \frac{\gamma_\theta}{2L}\mathbb{E}_r\left[\|F_{m_r}(\theta^r)\|^2\right]$$
$$\quad + \frac{\gamma_\theta}{2L}\mathbb{E}_r\left[\|F_{m_r}(\theta^r)\|^2\right] + 2\gamma_\theta\langle F(\theta^r),\theta^r-\hat{\theta}^r\rangle$$
$$= \|\hat{\theta}^r-\theta^*\|^2 - \frac{3\gamma_\theta}{2}\langle F(\theta^r),\theta^r-\theta^*\rangle + 2\gamma_\theta\langle F(\theta^r),\theta^r-\hat{\theta}^r\rangle$$
$$= \|\hat{\theta}^r-\theta^*\|^2 - \gamma_\theta\langle F(\theta^r),\theta^r-\theta^*\rangle - \frac{\gamma_\theta}{2}\langle F(\theta^r),\theta^r-\theta^*\rangle$$
$$\quad + 2\gamma_\theta\langle F(\theta^r),\theta^r-\hat{\theta}^r\rangle$$
$$\overset{(ii)}{\leq} \|\hat{\theta}^r-\theta^*\|^2 - \frac{\gamma_\theta}{L}\|F(\theta^r)\|^2 - \frac{\gamma_\theta}{2}\langle F(\theta^r),\theta^r-\theta^*\rangle + \frac{\gamma_\theta}{2L}\|F(\theta^r)\|^2$$
$$\quad + 2L\gamma_\theta\|\theta^r-\hat{\theta}^r\|^2$$
$$= \|\hat{\theta}^r-\theta^*\|^2 - \frac{\gamma_\theta}{2L}\|F(\theta^r)\|^2 - \frac{\gamma_\theta}{2}\langle F(\theta^r),\theta^r-\theta^*\rangle + 2L\gamma_\theta\|\theta^r-\hat{\theta}^r\|^2.$$

where in $(i)$ we use Assumption 1 and stepsize restriction $\gamma_\theta \leq \frac{1}{2L}$; in $(ii)$ we use Assumption 1 and Cauchy-Shwartz inequality. Rearranging the terms, we have

$$\frac{\gamma_\theta}{2L}\|F(\theta^r)\|^2 \leq \mathbb{E}_r\left[\|\hat{\theta}^{r+1}-\theta^*\|^2\right] - \|\hat{\theta}^r-\theta^*\|^2 - \frac{\gamma_\theta}{2}\langle F(\theta^r),\theta^r-\theta^*\rangle + 2L\gamma_\theta\|\theta^r-\hat{\theta}^r\|^2.$$

After averaging over iterations from $r=0$ to $R-1$ we get

$$\frac{\gamma_\theta}{2L}\frac{1}{R}\sum_{r=0}^{R-1}\mathbb{E}\left[\|F(\theta^r)\|^2\right] \leq \frac{\|\hat{\theta}^0-\theta^*\|^2}{R} + \frac{2L\gamma_\theta}{R}\sum_{r=0}^{R-1}\mathbb{E}\left[\|\theta^r-\hat{\theta}^r\|^2\right] - \frac{\gamma_\theta}{2R}\sum_{r=0}^{R-1}\mathbb{E}\left[\langle F(\theta^r),\theta^r-\theta^*\rangle\right].$$

Now we need to upper bound the third term. Using (16), we have

$$\sum_{r=0}^{R-1}\mathbb{E}\left[\|\theta^r-\hat{\theta}^r\|^2\right] = \sum_{r=0}^{R-1}\gamma_\theta^2\mathbb{E}\left[\left\|\sum_{m\in C^r}F_m(\theta^{\mathrm{prev}(m,r)})\right\|^2\right]$$
$$\leq M\gamma_\theta^2\sum_{r=0}^{R-1}\sum_{m\in C^r}\mathbb{E}\left[\|F_m(\theta^{\mathrm{prev}(m,r)})\|^2\right]$$
$$\overset{\mathrm{As.}1}{\leq} \gamma_\theta^2 ML\sum_{r=0}^{R-1}\sum_{m\in C^r}\mathbb{E}\left[\langle F_m(\theta^{\mathrm{prev}(m,r)}),\theta^{\mathrm{prev}(m,r)}-\theta^*\rangle\right]$$
$$= \gamma_\theta^2 ML\sum_{r=0}^{R-1}\sum_{m\in C^r}\mathbb{E}\left[\langle F(\theta^{\mathrm{prev}(m,r)}),\theta^{\mathrm{prev}(m,r)}-\theta^*\rangle\right].$$

The term $\mathbb{E}\left[\langle F(\theta^{\mathrm{prev}(m,r)}),\theta^{\mathrm{prev}(m,r)}-\theta^*\rangle\right]$ appears in the right hand side $\tau_{\max}$ times at most. Indeed, it appears for all iterations between $\mathrm{prev}(m,r)$ and $r$, which is upper bounded by $\tau_{\max}$. Thus, we have

$$\sum_{r=0}^{R-1}\mathbb{E}\left[\|\theta^r-\hat{\theta}^r\|^2\right] \leq \gamma_\theta^2 ML\tau_{\max}\sum_{r=0}^{R-1}\mathbb{E}\left[\langle F(\theta^r),\theta^r-\theta^*\rangle\right].$$

If $\gamma_\theta \leq \frac{1}{2L\sqrt{2M\tau_{\max}}}$, then we have

$$\sum_{r=0}^{R-1} \mathbb{E}\left[\|\theta^r - \hat{\theta}^r\|^2\right] \leq \frac{1}{8L} \sum_{r=0}^{R-1} \mathbb{E}\left[\langle F(\theta^r), \theta^r - \theta^*\rangle\right].$$

Thus, we derive

$$\frac{\gamma_\theta}{2L} \frac{1}{R} \sum_{r=0}^{R-1} \mathbb{E}\left[\|F(\theta^r)\|^2\right] \leq \frac{\|\hat{\theta}^0 - \theta^*\|^2}{R} + \frac{2L\gamma_\theta}{R} \frac{1}{8L} \sum_{r=0}^{R-1} \mathbb{E}\left[\langle F(\theta^r), \theta^r - \theta^*\rangle\right]$$

$$- \frac{\gamma_\theta}{2R} \sum_{r=0}^{R-1} \mathbb{E}\left[\langle F(\theta^r), \theta^r - \theta^*\rangle\right]$$

$$\leq \frac{\|\hat{\theta}^0 - \theta^*\|^2}{R}.$$

Finally, we get

$$\min_{r<R} \mathbb{E}\left[\|F(\theta^r)\|^2\right] \leq \frac{2L\|\hat{\theta}^0 - \theta^*\|^2}{\gamma_\theta R}.$$

$\square$

**Remark 3.** We highlight the fact that if we use the stepsize $\gamma_\theta = \frac{1}{2L\sqrt{2M\tau_{\max}}}$, then the convergence is

$$\min_{r<R} \mathbb{E}\left[\|F(\theta^r)\|^2\right] \leq \frac{4L^2\sqrt{2M\tau_{\max}}\|\hat{\theta}^0 - \theta^*\|^2}{R}.$$

We observe a square root dependency on $\tau_{\max}$. The same result has been recently derived in a homogeneous setting (Koloskova et al., 2022) for vanilla Asynchronous SGD.

# D   Byzantine-robust version

**Preliminaries.**   We assume that among $M$ clients participating in the training, there is a subset of clients $\mathcal{B}$ called *Byzantines*, i.e., clients that can (intentionally or not) deviate from the prescribed algorithm and are omniscient (i.e., they know the updates of other clients and aggregation rule applied by the server). More precisely, we assume that $[M] = \mathcal{G} \sqcup \mathcal{B}$, where $\mathcal{G}$ denotes the set of regular clients, $|\mathcal{G}| := G$, $|\mathcal{B}| := B \leq \delta M$, where $\delta < 1/2$. The goal is to solve an instance of (2) with

$$F(\theta) = \frac{1}{G} \sum_{m \in \mathcal{G}} F_m(\theta), \tag{17}$$

where operators $\{F_m\}_{m \in \mathcal{G}}$ are defined as $F_m(\theta) = \nabla_1 f_m(\theta, w_m^*(\theta))$.

Since the standard averaging is vulnerable to Byzantine attacks, we use robust aggregation rules in the sense of the following definition.

**Definition 2** $((\delta, c)$-robust aggregator (Karimireddy et al., 2021; 2022; Gorbunov et al., 2023)). Let random vectors $\{x_1, \ldots, x_M\}$ are such that there exists a subset $\mathcal{G} \subseteq [M]$ such that $|\mathcal{G}| = G \geq (1 - \delta)n$ where $\delta < 1/2$ and for some $\sigma \geq 0$ the following inequality holds: $\frac{1}{G(G-1)} \sum_{m,n \in \mathcal{G}} \mathbb{E}[\|x_m - x_n\|^2] \leq \sigma^2$ where the expectation is taken w.r.t. the randomness of $\{x_m\}_{m \in \mathcal{G}}$. Then, vector $\hat{x}$ is called $(\delta, c)$-Robust Aggregator $((\delta, c) - \mathtt{RAgg})$ for some $c > 0$ and denoted as $\hat{x} = \mathtt{RAgg}(x_1, \ldots, x_M)$ if the following condition holds:

$$\mathbb{E}\left[\|\hat{x} - \overline{x}\|^2\right] \leq c\delta\sigma^2, \tag{18}$$

where $\overline{x} = \frac{1}{G} \sum_{m \in \mathcal{G}} x_i$. If, in addition, the computation of $\hat{x}$ is independent of $\sigma^2$, $\hat{x}$ is called $(\delta, c)$-Agnostic Robust Aggregator $((\delta, c) - \mathtt{ARAgg})$ and denoted as $\hat{x} = \mathtt{ARAgg}(x_1, \ldots, x_M)$.

---

**Algorithm 5** Byzantine-Robust Fine-tuning Followed by Global Gradient (BR-FFGG)

---

1: **Input:** initialization $\theta^0 \in \mathbb{R}^d$, stepsizes $\gamma_w, \gamma_\theta > 0$, $(\delta, c)$-`ARAgg`
2: **for** $r = 0, 1, 2, \dots$ **do**
3:     **for** client $m \in \mathcal{G}$ **do**
4:         Compute and send $g_m(\theta^r)$ – an unbiased estimate of $F_m(\theta^r)$
5:     **end for**
6:     **for** client $m \in \mathcal{B}$ **do**
7:         Send $g_m(\theta^r) = *$               $\triangleright$ Byzantine clients can send anything to the server
8:     **end for**
9:     $\theta^{r+1} = \theta^r - \gamma_\theta \texttt{ARAgg}(g_1(\theta^r), \dots, g_M(\theta^r))$
10: **end for**

---

This definition is tight in a certain sense (see the details in (Karimireddy et al., 2021)) and is not satisfied for such defenses as Krum (Blanchard et al., 2017), geometric median (Pillutla et al., 2022a), and coordinate-wise median (Chen et al., 2017) that are known to be insufficient to ensure Byzantine-robustness (Baruch et al., 2019; Xie et al., 2020). For the examples of aggregators satisfying Definition 2 we refer to Gorbunov et al. (2023).

We also make an additional assumption related to the overparameterization.

**Assumption 6.** *We assume that any regular client $m \in \mathcal{G}$ can compute an unbiased estimate $g_m(\theta)$ of $F_m(\theta)$, i.e., $\mathbb{E}[g_m(\theta)] = F_m(\theta)$, and for any $\theta \in \mathbb{R}^d$*

$$\mathbb{E}\left[\|g_m(\theta)\|^2\right] \le \rho_{in}\|F_m(\theta)\|^2. \tag{19}$$

*In addition, we assume that for any $\theta \in \mathbb{R}^d$ we have*

$$\frac{1}{G}\sum_{m \in \mathcal{G}} \|F_m(\theta)\|^2 - \|F(\theta)\|^2 \le \ell_{sim}\langle F(\theta), \theta - \theta^*\rangle. \tag{20}$$

For example, inequality (19) is satisfied with $\rho_{\text{in}} = 1$ when $g_m(\theta) = F_m(\theta)$, i.e., when regular clients compute $w_m(\theta^r)$ and $F_m(\theta^r)$ exactly at each step, and can be satisfied when the clients have overparameterized data (locally). Inequality (20) is satisfied with $\ell_{\text{sim}} \le L$ whenever Assumption 1 holds. However, $\ell_{\text{sim}}$ can be much smaller than $L$ if local operators $\{F_m\}_{m \in \mathcal{G}}$ are similar.

Finally, we made an extra assumption on structured non-monotonicity of operators $\{F_m\}_{m \in \mathcal{G}}$.

**Assumption 7.** *We assume that for all $m \in \mathcal{G}$ operators $F_m$ are $\mu$-quasi strongly monotone, i.e., for all $\theta \in \mathbb{R}^d$ and $\theta^*$ such that $F_m(\theta^*)$ we have*

$$\langle F_m(\theta), \theta - \theta^*\rangle \ge \mu\|\theta - \theta^*\|^2. \tag{21}$$

Standard strong monotonicity, i.e., $\langle F_m(\theta_1) - F_m(\theta_2), \theta_1 - \theta_2\rangle \ge \mu\|\theta_1 - \theta_2\|^2$ for any $\theta_1, \theta_2 \in \mathbb{R}^d$, implies condition from (21) (Mertikopoulos & Zhou, 2019; Song et al., 2020; Loizou et al., 2021) but the opposite implication is not always true. Moreover, as it is shown in (Loizou et al., 2021), an operator can be non-monotone but quasi-strongly monotone.

**Theorem 8.** *Let Assumptions 1, 2, 6, 7 hold. Assume that*

$$\delta \le \frac{\mu}{2c\left(\left(\rho_{in} + \frac{1}{G-1}\right)\ell_{sim} + (\rho_{in} - 1)L\right)}, \tag{22}$$

$$\gamma_\theta \le \frac{1}{4\left(\frac{(\rho_{in}-1)(L+\ell_{sim})}{G} + L + 2c\delta\left(\left(\rho_{in} + \frac{1}{G-1}\right)\ell_{sim} + (\rho_{in} - 1)L\right)\right)}. \tag{23}$$

*Then, for any $R \ge 0$ the iterates produced by Algorithm 5 satisfy*

$$\mathbb{E}\left[\|\theta^R - \theta^*\|^2\right] \le \left(1 - \frac{\gamma_\theta\mu}{2}\right)^R \|\theta^0 - \theta^*\|^2. \tag{24}$$

*Proof.* To simplify the derivation, we introduce new vectors: $\widehat{g}^r = \texttt{ARAgg}(g_1(\theta^r), \ldots, g_M(\theta^r))$ and $\overline{g}^r = \frac{1}{G}\sum_{m\in\mathcal{G}} g_m(\theta^r)$. Then, $\theta^{r+1} = \theta^r - \gamma_\theta \widehat{g}^r$ and

$$
\begin{aligned}
\|\theta^{r+1} - \theta^*\|^2 &= \|\theta^r - \theta^*\|^2 - 2\gamma_\theta \langle \widehat{g}^r, \theta^r - \theta^* \rangle + \gamma_\theta^2 \|\widehat{g}^r\|^2 \\
&\leq \|\theta^r - \theta^*\|^2 - 2\gamma_\theta \langle \overline{g}^r, \theta^r - \theta^* \rangle - 2\gamma_\theta \langle \widehat{g}^r - \overline{g}^r, \theta^r - \theta^* \rangle \\
&\quad + 2\gamma_\theta^2 \|\overline{g}^r\|^2 + 2\gamma_\theta^2 \|\widehat{g}^r - \overline{g}^r\|^2 \\
&\leq \left(1 + \frac{\gamma_\theta \mu}{2}\right) \|\theta^r - \theta^*\|^2 - 2\gamma_\theta \langle \overline{g}^r, \theta^r - \theta^* \rangle + 2\gamma_\theta^2 \|\overline{g}^r\|^2 \\
&\quad + \gamma_\theta \left(2\gamma_\theta + \frac{1}{2\mu}\right) \|\widehat{g}^r - \overline{g}^r\|^2,
\end{aligned}
$$

where in the second step we use $\|a+b\|^2 \leq 2\|a\|^2 + 2\|b\|^2$ and, in the last step, we apply $\langle a, b \rangle \leq \frac{\alpha}{2}\|a\|^2 + \frac{1}{2\alpha}\|b\|^2$ that hold for any $a, b \in \mathbb{R}^d$ and $\alpha > 0$. Taking the expectation $\mathbb{E}_r[\cdot]$ w.r.t. the randomness coming from the $r$-th step and using $\mathbb{E}_r[\overline{g}^r] = F(\theta^r)$, we derive

$$
\begin{aligned}
\mathbb{E}_r\left[\|\theta^{r+1} - \theta^*\|^2\right] &\leq \left(1 + \frac{\gamma_\theta \mu}{2}\right) \|\theta^r - \theta^*\|^2 - 2\gamma_\theta \langle F(\theta^r), \theta^r - \theta^* \rangle + 2\gamma_\theta^2 \mathbb{E}_r\left[\|\overline{g}^r\|^2\right] \\
&\quad + \gamma_\theta \left(2\gamma_\theta + \frac{1}{2\mu}\right) \mathbb{E}_r\left[\|\widehat{g}^r - \overline{g}^r\|^2\right].
\end{aligned}
\tag{25}
$$

Next, we use independence of $\{g_m(\theta^r)\}_{m\in\mathcal{G}}$:

$$
\begin{aligned}
\mathbb{E}_r\left[\|\overline{g}^r\|^2\right] &= \mathbb{E}_r\left[\|\overline{g}^r - F(\theta^r)\|^2\right] + \|F(\theta^r)\|^2 \\
&= \frac{1}{G^2}\sum_{m\in\mathcal{G}} \mathbb{E}_r\left[\|g_m(\theta^r) - F_m(\theta^r)\|^2\right] + \|F(\theta^r)\|^2 \\
&\overset{(19)}{\leq} \frac{\rho_{\text{in}} - 1}{G^2}\sum_{m\in\mathcal{G}} \|F_m(\theta^r)\|^2 + \|F(\theta^r)\|^2 \\
&= \frac{\rho_{\text{in}} - 1}{G}\left(\frac{1}{G}\sum_{m\in\mathcal{G}} \|F_m(\theta^r)\|^2 - \|F(\theta^r)\|^2\right) + \left(1 + \frac{\rho_{\text{in}} - 1}{G}\right)\|F(\theta^r)\|^2 \\
&\overset{(4),(20)}{\leq} \left(\frac{(\rho_{\text{in}} - 1)(L + \ell_{\text{sim}})}{G} + L\right)\langle F(\theta^r), \theta^r - \theta^* \rangle.
\end{aligned}
\tag{26}
$$

To upper-bound $\mathbb{E}_r\left[\|\widehat{g}^r - \overline{g}^r\|^2\right]$, we need to estimate $\text{PV}_r := \frac{1}{G(G-1)}\sum_{m,n\in\mathcal{G}} \mathbb{E}_r\left[\|g_m(\theta^r) - g_n(\theta^r)\|^2\right]$:

$$
\begin{aligned}
\text{PV}_r &= \frac{1}{G(G-1)}\sum_{\substack{m,n\in\mathcal{G} \\ m\neq n}} \mathbb{E}_r\left[\|g_m(\theta^r)\|^2 + \|g_n(\theta^r)\|^2\right] - \frac{2}{G(G-1)}\sum_{\substack{m,n\in\mathcal{G} \\ m\neq n}} \langle F_m(\theta^r), F_n(\theta^r) \rangle \\
&= \frac{2}{G}\sum_{m\in\mathcal{G}} \mathbb{E}_r\left[\|g_m(\theta^r)\|^2\right] - \frac{2}{G-1}\sum_{m\in\mathcal{G}} \left\langle F_m(\theta^r), \frac{1}{G}\sum_{\substack{n\in\mathcal{G} \\ n\neq m}} F_n(\theta^r) \right\rangle \\
&= \frac{2}{G}\sum_{m\in\mathcal{G}} \mathbb{E}_r\left[\|g_m(\theta^r)\|^2\right] + \frac{2}{G(G-1)}\sum_{m\in\mathcal{G}} \|F_m(\theta^r)\|^2 - \frac{2}{G-1}\sum_{m\in\mathcal{G}} \langle F_m(\theta^r), F(\theta^r) \rangle \\
&\overset{(19)}{\leq} \frac{2((G-1)\rho_{\text{in}} + 1)}{G(G-1)}\sum_{m\in\mathcal{G}} \|F_m(\theta^r)\|^2 - \frac{2G}{G-1}\|F(\theta^r)\|^2 \\
&= \frac{2((G-1)\rho_{\text{in}} + 1)}{G-1}\left(\frac{1}{G}\sum_{m\in\mathcal{G}} \|F_m(\theta^r)\|^2 - \|F(\theta^r)\|^2\right) + 2(\rho_{\text{in}} - 1)\|F(\theta^r)\|^2 \\
&\overset{(20),(4)}{\leq} 2\left(\left(\rho_{\text{in}} + \frac{1}{G-1}\right)\ell_{\text{sim}} + (\rho_{\text{in}} - 1)L\right)\langle F(\theta^r), \theta^r - \theta^* \rangle.
\end{aligned}
\tag{27}
$$

In view of Definition 2, this upper bound gives us

$$\mathbb{E}_r\left[\|\widehat{g}^r - \overline{g}^r\|^2\right] \le c\delta \mathrm{PV}_r \overset{(27)}{\le} 2c\delta\left(\left(\rho_{\mathrm{in}} + \frac{1}{G-1}\right)\ell_{\mathrm{sim}} + (\rho_{\mathrm{in}} - 1)L\right)\langle F(\theta^r), \theta^r - \theta^*\rangle. \qquad (28)$$

Plugging (26) and (28) in (25), we get

$$
\begin{aligned}
\mathbb{E}_r\left[\|\theta^{r+1} - \theta^*\|^2\right] &\le \left(1 + \frac{\gamma_\theta\mu}{2}\right)\|\theta^r - \theta^*\|^2 \\
&\quad - 2\gamma_\theta\left(1 - \gamma_\theta\left(\frac{(\rho_{\mathrm{in}}-1)(L+\ell_{\mathrm{sim}})}{G} + L\right)\right)\langle F(\theta^r), \theta^r - \theta^*\rangle \qquad (29) \\
&\quad + 2c\delta\gamma_\theta\left(2\gamma_\theta + \frac{1}{2\mu}\right)\left(\left(\rho_{\mathrm{in}} + \frac{1}{G-1}\right)\ell_{\mathrm{sim}} + (\rho_{\mathrm{in}}-1)L\right)\langle F(\theta^r), \theta^r - \theta^*\rangle \\
&\overset{(22)}{\le} \left(1 + \frac{\gamma_\theta\mu}{2}\right)\|\theta^r - \theta^*\|^2 \\
&\quad - 2\gamma_\theta\left(\frac{3}{4} - \gamma_\theta\left(\frac{(\rho_{\mathrm{in}}-1)(L+\ell_{\mathrm{sim}})}{G} + L\right)\right)\langle F(\theta^r), \theta^r - \theta^*\rangle \\
&\quad + 4c\delta\gamma_\theta^2\left(\left(\rho_{\mathrm{in}} + \frac{1}{G-1}\right)\ell_{\mathrm{sim}} + (\rho_{\mathrm{in}}-1)L\right)\langle F(\theta^r), \theta^r - \theta^*\rangle \\
&\overset{(23)}{\le} \left(1 + \frac{\gamma_\theta\mu}{2}\right)\|\theta^r - \theta^*\|^2 - \gamma_\theta\langle F(\theta^r), \theta^r - \theta^*\rangle \\
&\overset{(21)}{\le} \left(1 - \frac{\gamma_\theta\mu}{2}\right)\|\theta^r - \theta^*\|^2.
\end{aligned}
$$

Taking the full expectation from the above inequality and unrolling the recurrence, we obtain the result. □

The derived result establishes linear convergence to the exact solution (asymptotically, in expectation) with the possible presence of Byzantine clients. For simplicity, let us consider the case when $\rho_{\mathrm{in}} = 1$. As mentioned earlier, this case corresponds to the exact computation of $F_m(\theta^r)$ for all $m \in \mathcal{G}$. Then, conditions (22) and (23) reduce to

$$\delta \le \frac{\mu}{2c\left(\left(1 + \frac{1}{G-1}\right)\ell_{\mathrm{sim}}\right)}, \quad \gamma_\theta \le \frac{1}{4\left(L + 2c\delta\left(1 + \frac{1}{G-1}\right)\ell_{\mathrm{sim}}\right)}.$$

In the worst case, $\ell_{\mathrm{sim}} = L$ that can be much larger than $\mu$ and, thus, implies that $\delta$ should be very small for the derived result. In the context of minimization, similar pathological behavior is observed in (Karimireddy et al., 2022; Gorbunov et al., 2023). In particular, the existing SOTA theoretical results under the assumption $\frac{1}{G}\sum_{m\in\mathcal{G}}\|\nabla f_m(\theta,w)\|^2 \le B^2\|\nabla f(\theta,w)\|^2$ (Karimireddy et al., 2022; Gorbunov et al., 2023) require $\delta \lesssim 1/cB^2$. However, when all functions $\{f_m\}_{m\in\mathcal{G}}$ are $L$-smooth, have shared minimum, and $f$ is $\mu$-strongly convex, then, in the worst case, $B^2 = L/\mu$. Up to numerical constants and the differences between definitions of $L$ and $\mu$ in our work and in (Karimireddy et al., 2022; Gorbunov et al., 2023), we get the same worst-case upper-bound for $\delta$.

However, when $\ell_{\mathrm{sim}} \ll L$, our condition on $\delta$ can be very mild. For example, when the data on workers is similar to a certain extent, then $\ell_{\mathrm{sim}}$ can be of the order of $\mu$ or even smaller. In this case, our condition on $\delta$ can become void, and the upper bound for $\delta$ will be determined by the type of aggregation rule (see the examples in (Karimireddy et al., 2022)).

# E   Experiments

## E.1   How does data heterogeneity affect the convergence?

In this section[1], we provide the experiments on Example 1 to demonstrate the robustness of the proposed FFGG algorithm varying the data heterogeneity of the optimization problem. To do so, we follow Section 5.1

---

[1]Our implementation for this section is available at https://github.com/Rustem-Islamov/FL_representations.

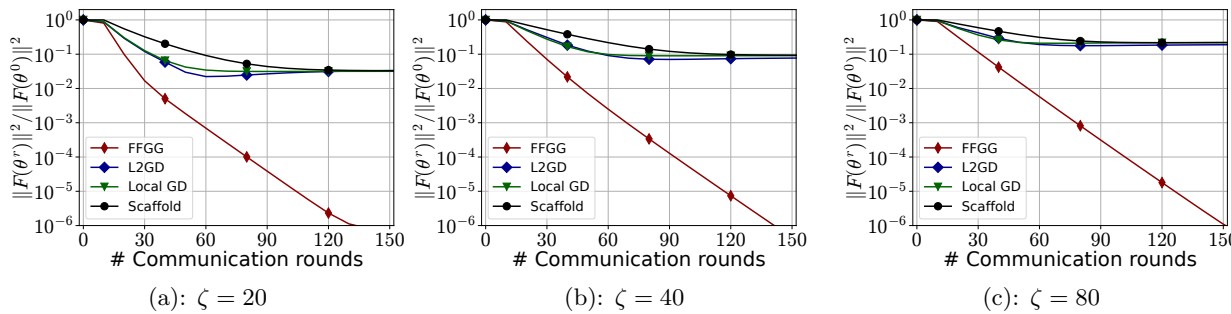

Figure 2: Comparison of FFGG against Scaffold, Local GD, and L2GD varying the data heterogeneity constant $\zeta$. FFGG uses gradient descent as a subsolver with the same number of local steps as the other methods to make the comparison fair.

and construct the matrices $\mathbf{H}_m, \mathbf{A}_m, \mathbf{B}_m$ and vectors $y_m, b_m$ as follows. First, we sample matrices $\mathbf{H}, \mathbf{A}, \in \mathbb{R}^{n \times d_\theta}$ and $\mathbf{B} \in \mathbb{R}^{d \times d_w}$ from standard uniform distribution and rescale by the second dimension, i.e.

$$\mathbf{H}, \mathbf{A} \sim \frac{1}{d_\theta}[\mathrm{Unif}(0,1)]^{n \times d_\theta}, \quad \mathbf{B} \sim \frac{1}{d_w}[\mathrm{Unif}(0,1)]^{n \times d_w}.$$

Second, we choose a number $\zeta > 0$ and for each client $m \in [M]$ we define the matrices

$$\mathbf{H}_m = \mathbf{H} + \zeta \cdot \frac{\mathbf{H}'_m}{\|\mathbf{H}'_m\|_2}, \quad \mathbf{A}_m = \mathbf{A} + \zeta \cdot \frac{\mathbf{A}'_m}{\|\mathbf{A}'_m\|_2}, \quad \mathbf{B}_m = \mathbf{B} + \zeta \cdot \frac{\mathbf{B}'_m}{\|\mathbf{B}'_m\|_2},$$

where

$$\mathbf{H}'_m, \mathbf{A}'_m \sim [\mathcal{N}(0,1)]^{n \times d_\theta}, \quad \mathbf{B}'_m \sim [\mathcal{N}(0,1)]^{n \times d_w}.$$

Here the constant $\zeta$ controls the heterogeneity of the problem. The higher the value of $\zeta$, the more heterogeneous the problem becomes. Finally, the vectors $y_m$ and $b_m$ are constructed by

$$y_m = \mathbf{A}_m y_1 + \mathbf{B}_m y_2 + \iota y_3, \quad \text{where} \quad y_1 \sim [\mathcal{N}(0,1)]^{d_\theta}, y_2 \sim [\mathcal{N}(0,1)]^{d_w}, y_3 \sim [\mathcal{N}(0,1)]^n,$$

$$b_m = \mathbf{H}_m b_1 + \iota b_2 \quad \text{where} \quad b_1 \sim [\mathcal{N}(0,1)]^{d_\theta}, b_2 \sim [\mathcal{N}(0,1)]^n,$$

and $\iota = 10^{-3}$. We set $d_\theta = 100, d_w = 50, n = 1000, M = 32$, and vary $\zeta \in \{20, 40, 80\}$. We consider the same set of algorithms as in Section 5.1 and set the theoretical values of the stepsize for all of them. All algorithms use local gradient updates with $\tau = 20$ local steps.

The results are reported in Figure 2. We observe that in all the cases FFGG algorithm achieves the gradient norm of order $10^{-6}$ after about $130 - 150$ communication rounds, i.e. the change in data heterogeneity has negligible impact on the convergence of FFGG. In contrast, the other three algorithms are affected more since the gradient norm that they achieve gets worse with $\zeta$. These results demonstrate the stability of FFGG in relation to the variations in the data heterogeneity of the problem and support the theoretical findings of our work.

### E.2 The convergence of asynchronous FFGG

Next, we demonstrate the convergence of the asynchronous version of FFGG summarized in Algorithm 3. We follow the experimental setup of (Mishchenko et al., 2022a) and test both synchronous and asynchronous versions of FFGG on the same problem as in the previous section. We set $M = 40, d_\theta = 400, d_w = 50, n = 1000$, and $\zeta = 10$. Both versions use $\tau = \{5, 10, 50, 200\}$ local steps of GD. We use the Ray package (Moritz et al., 2018) to parallelize the execution and follow the official documentation for the implementation[2] of

---

[2] https://docs.ray.io/en/latest/ray-core/examples/plot_parameter_server.html#asynchronous-parameter-server-training

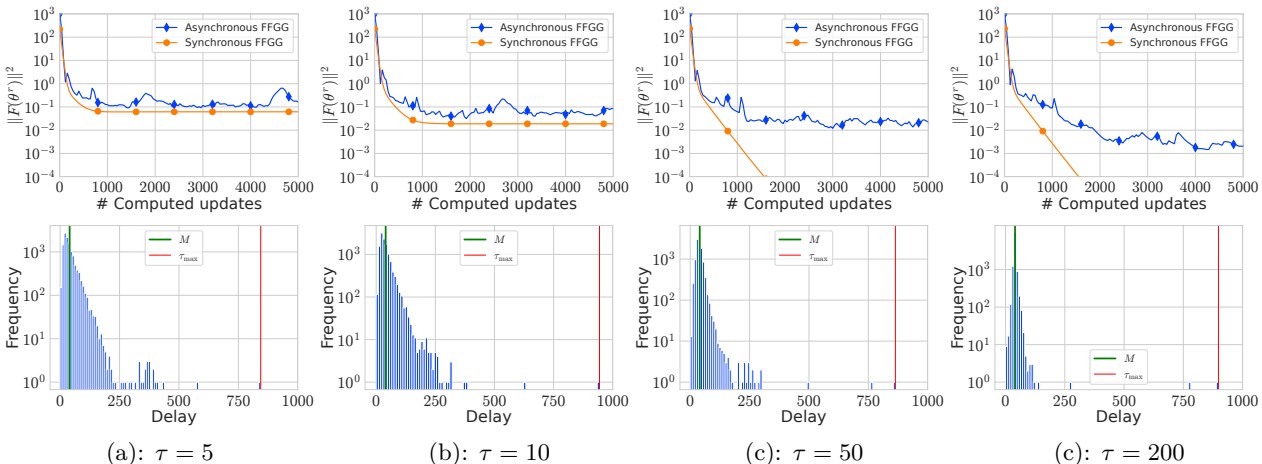

Figure 3: **First line:** Comparison of asynchronous and synchronous variants of FFGG in terms of the number of computed update directions based on the local GD. **Second line:** Delays of updates using asynchronous FFGG.

asynchronous training. All delays appearing in the runs come from the execution on CPUs in contrast to previous works where the delays are simulated according to some distribution (Koloskova et al., 2022; Islamov et al., 2024). We tune both the inner and outer stepsize of FFGG by setting the stepsizes to theoretical values multiplied by a factor $\{0.25, 0.5, 1, 2, 4\}$.

We demonstrate the results in Figure 3 (first line) varying the number of steps $\tau$ of Local GD. We observe that the synchronous variant is faster than the asynchronous one. Moreover, the asynchronous FFGG improves by increasing the local steps $\tau$ similar to the synchronous FFGG. However, the improvement is slower than that of the synchronous variant. We report the delays throughout the optimization process in Figure 3 (second line). They suggest that the maximum delay is typically much larger than the number of workers in the system.

### E.3 The convergence of FFGG with Byzantine workers

In our next experiment, we demonstrate the performance of the Byzantine-robust version of FFGG summarized in Algorithm 5. To do so, we consider the same data generation mechanism as in Section 5.1 with $n = 10000, d_\theta = 100, d_w = 50$, and $M = 42$. To demonstrate the performance, we use the exact computations on the worker side, i.e., each worker $m$ computes locally $\operatorname{argmin}_w f_m(\theta^{r+1}, w)$. We use coordinate-wise median with bucketing as an aggregation mechanism on the server (Chen et al., 2017; Karimireddy et al., 2020a). The 10 of all workers are Byzantine (which corresponds to $\approx 25\%$ of all workers), and they flip the sign of the gradient. For each algorithm, we use the theoretical values for the stepsizes. We report the convergence results in Figure 4. According to the plot, even in the presence of Byzantine workers FFGG algorithm still convergences linearly fast to the optimum. The convergence speed slightly slows down when there are Byzantine workers in training in comparison to non-Byzantine training as expected.

### E.4 Expertimental details for real-world datasets

As mentioned in the main part, we adhere to the experimental arrangement outlined in (Pillutla et al., 2022b) for consistency. To provide comprehensive information, we present the setup below.

Our experiments are conducted using two datasets encompassing two modalities, specifically images, and text. These datasets feature a natural division of data that is non-i.i.d., mirroring the heterogeneity of data encountered in real-world Federated Learning scenarios. We provide a detailed account of the experimental

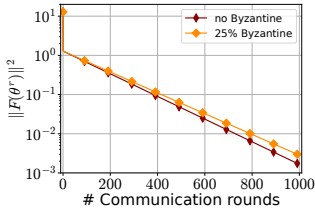

Figure 4: The performance of FFGG algorithm with 25% of workers being Byzantine and without Byzantine workers.

setup and hyperparameters employed. We base our implementation on the publicly available code provided by Pillutla et al. (2022) [3].

### E.5  Datasets, tasks and models

We explore two tasks inspired by real-world applications of Federated Learning: StackOverflow for next-word prediction and EMNIST for character recognition.

As part of our approach to partial personalization, we consider three partitioning schemes:

- *Input-layer personalization*: This architectural design focuses on customizing the input layer to learn personalized representations while the remaining parts of the model are shared among all clients. Specifically, in the case of predicting the next word, we personalize the initial transformer layer instead of the embedding layer.

- *Output-layer personalization*: With this design, we aim to learn a shared representation while customizing the prediction layer. In the case of a transformer model, we adapt the final transformer layer instead of the output layer to achieve personalization.

- *Adapter personalization*: In this scheme, each client utilizes a personalized low-rank adapter to fine-tune the global model.

These partitioning schemes serve as strategies for incorporating partial personalization into Federated Learning, allowing for different levels of customization while leveraging a shared model across clients.

### E.5.1  StackOverflow for next word prediction

**Dataset.**  The dataset used for our task is derived from Stack Overflow, a popular programming question-answer website. It consists of questions and corresponding answers. In the next word prediction task, the objective is to forecast the subsequent word based on a partial sequence of words within a question or answer. This particular task serves as a valuable open-source benchmark for evaluating next-word prediction capabilities in mobile keyboards. For our experiments, we utilize the StackOverflow dataset made available by TensorFlow Federated[4].

**Client distributions.**  Each client in our study corresponds to an individual user on Stack Overflow, and the data available to each client comprises the questions and answers posted by that specific user. To ensure an adequate amount of data for analysis, we only include clients with a minimum of 100 training sequences and 10 testing sequences. Here, a sequence refers to either a question or an answer posted by the user. We further narrow down the dataset by utilizing a fixed subsample of 1000 clients.

Consistent with the approach described in (Reddi et al., 2020), we limit the vocabulary to the top 10000 most frequently occurring words in the dataset. Additionally, we apply padding and truncation techniques

---

[3]https://github.com/facebookresearch/FL_partial_personalization
[4]https://www.tensorflow.org/federated

to standardize the length of each sequence within each client, setting it to 20. Furthermore, we consider a maximum of 1000 training sequences per client during our analysis.

**Model.** For our implementation, we employ a transformer model (Vaswani et al., 2017) that is similar in size to BERT Mini (Turc et al., 2019). The model consists of 4 transformer blocks, and each self-attention layer is equipped with 4 attention heads. The transformer hidden dimension is set to 256, while the fully connected hidden dimension is 1024.

The model incorporates a causal language modeling head, which refers to a fully connected layer responsible for assigning scores to all possible vocabulary items, including special tokens.

**Loss function and evaluation metric.** During the training phase, we utilize the causal language modeling objective. This means that for each partial sequence, we treat the task of predicting the next word as a multiclass classification problem and aim to minimize the cross-entropy loss.

For evaluation purposes, we employ the top-1 accuracy metric, which measures the accuracy of predicting the correct word from the proper 10000-word vocabulary. This evaluation metric disregards special tokens such as padding, out-of-vocabulary terms, and beginning/end of sequence markers.

### E.5.2 GLDv2 for visual landmark recognition

**Dataset.** Google Landmarks Dataset v2 (GLDv2) (Weyand et al., 2020) is a large-scale image dataset. This dataset comprises pictures of well-known landmarks across the globe, all of which were captured and uploaded by contributors to Wikipedia. Although the image dimensions vary, the most prevalent size is 800 by 600 pixels.

The primary objective of the visual landmark recognition assignment is to pinpoint the landmark depicted in the image. This task mirrors real-life situations where individuals use their smartphones to take pictures of natural or architectural landmarks during their travels. We utilize the federated version of the GLDv2 dataset by (Hsu et al., 2020), which includes 2028 landmarks and is provided by TensorFlow Federated.

**Client distributions.** Every client is associated with a specific Wikipedia user and includes all images contributed by that user. We only incorporate the 823 clients that have a minimum of 50 datapoints. We don't utilize the original test set from GLDv2 for evaluation, as it originates from distinct clients. Rather, we allocate 50% of the data from each client to be used as a test set.

**Model.** We employ a ResNet-18 (He et al., 2016) model, which has been pretrained on the ImageNet dataset (Deng et al., 2009). We use group normalization in place of batch normalization. All images are resized to dimensions of 224 by 224 pixels. Our training incorporates two data augmentations: a random cropping to a 256 by 256 size and a random horizontal flip.

**Loss function and evaluation metric.** We use the cross-entropy loss. The model's effectiveness is evaluated based on its classification accuracy.

### E.5.3 EMNIST for character recognition

**Dataset.** The EMNIST dataset (Cohen et al., 2017) serves as a character recognition dataset. The objective is to identify images containing handwritten digits or letters, with a total of 62 possible options encompassing lowercase and uppercase letters (a-z, A-Z) as well as digits (0-9).

The images within the dataset are grayscale and have dimensions of $28 \times 28$, resulting in a total of 784 pixels. For our experiments, we utilize the EMNIST dataset made available by TensorFlow Federated.

**Client distributions.** In our study, each client represents an individual "writer," referring to the human subject who contributed by hand-writing the digit or letter during the data collection phase. We specifically consider clients that have a minimum of 100 training points and 25 testing points, resulting in a total of 1114 eligible clients for analysis.

Table 3: Hyperparameters for each dataset/task.

| | Hyperparameter | StackOverflow | GLDv2 | EMNIST |
|---|---|---|---|---|
| | Batch size | 64 | 64 | 32 |
| | Devices per round | 50 | 50 | 10 |
| | Local epochs | 1 | 1 | 1 |
| | Server optimizer | FedAdam | FedAdam | FedAvg |
| Common | Client optimizer | SGD | SGD | SGD |
| | Global scheduler | Linear | Linear | Exponential |
| | Warm-up | 10% of rounds | 10% of rounds | N/A |
| | LR decay rounds | N/A | N/A | |
| | Max. grad. norm. | 0.1 | N/A | N/A |
| Non-personalized training (step 1. of the pipeline) | # Rounds | 1000 | 2500 | 2000 |
| | Server learning rate | $5 \times 10^{-4}$ | $2 \times 10^{-4}$ | 1.0 |
| | Client learning rate | 1 | $10^{-2}$ | 0.5 |
| Personalized training (step 2. of the pipeline) | # Rounds | 500 | 600 | 500 |
| | Server learning rate | $5 \times 10^{-5}$ | $2 \times 10^{-5}$ | 1.0 |
| | Client learning rate | $10^{-1}$ | $10^{-3}$ | $10^{-2}$ |
| Local finetuning (step 3. of the pipeline) | #Epochs | 5 | 5 | 5 |
| | Optimizer | SGD | SGD | SGD |
| | Client learning rate | $10^{-1}$ | $10^{-3}$ | $10^{-2}$ |

**Model.** To address the smaller image size $(28 \times 28 \times 1)$ in our dataset, which differs from the $224 \times 224 \times 3$ size that the original ResNet was designed for, we utilize a ResNet-18 model (He et al., 2016). However, we make two modifications to accommodate this smaller size.

Firstly, we adjust the convolutional kernel size in the first convolution layer from the original $7 \times 7$ to $3 \times 3$. This modification allows the model to process the input images appropriately.

Secondly, we omit the first pooling layer that is present in the original ResNet architecture. By removing this layer, we ensure compatibility with our image size and maintain the effectiveness of the model for our specific task.

**Loss function and evaluation metric.** We use the cross-entropy loss. We evaluate the performance of the model using its classification accuracy.

### E.6 Experimental pipeline and baselines

There are three components in the training pipeline for all experiments:

1. **Non-personalized federated pre-training**: The first step involves training a global model without any personalization using FedAvg that we use to initialize (1).

2. **Partially personalized federated training**: This is the main training step that we describe in detail in Section 5.3.

3. **Final finetuning**: The last step involves only finetuning the personalized parameter $w_m^*(\theta)$ for each client.

### E.7 Hyperparameters and evaluation details

The hyperparameters we use are given in Table 3.

**Evaluation metric.** Our main evaluation metric for both next-word prediction and image classification tasks is the weighted average of test accuracy across all clients. This weighted average takes into account the number of test examples from each client, allowing for a comprehensive assessment of model performance. This evaluation metric is equivalent to an unweighted accuracy achieved by aggregating all the data centrally.

**Rounds.** We utilize the concept of communication rounds, which refers to the number of iterations during which the shared parameters are securely aggregated, to track the progress of each algorithm. In the case of non-personalized training, we set the number of rounds to 1000 for StackOverflow, 2000 for EMNIST, and 2500 rounds for GLDv2.

For personalized training, we initialize the model with the parameters obtained from the non-personalized training and continue training for an additional 500 rounds for both StackOverflow and EMNIST datasets, and 600 rounds for GLDv2.

**Devices per round.** We assume that all devices are accessible and selections are made in a uniformly random manner. Consistent with the approach described in (Reddi et al., 2020), we choose 50 devices per round for StackOverflow/GLDv2 and 10 devices per round for EMNIST. This selection process applies to both non-personalized and personalized training for all datasets.

**Local updates and minibatch size.** In both non-personalized and personalized federated training, each selected device executes one epoch of mini-batch stochastic gradient descent locally. Following this, during the final fine-tuning stage of personalized training, we perform five epochs of training.

For the StackOverflow and GLDv2 datasets, we utilize a mini-batch size of 64 across all settings. As for the EMNIST dataset, the mini-batch size is set to 32 for all configurations.

**Server and client optimizer details.** For the EMNIST dataset, we utilize the FedAvg algorithm, while for the StackOverflow dataset and GLDv2, we employ FedAdam (Reddi et al., 2020). Additionally, we incorporate a global scheduler that applies a schedule to the client learning rates across multiple rounds while maintaining a constant learning rate for each client within a round.

There are two types of schedulers we use: a linear scheduler and an exponential scheduler (referred to as "stepLR" in PyTorch). The linear scheduler involves a linear warmup, if applicable, until reaching the maximum learning rate, followed by a linear decay to 0. On the other hand, the exponential scheduler reduces the client learning rate by half after a fixed number of rounds.

