# OpenReview forum: "Partially Personalized Federated Learning: Breaking the Curse of Data Heterogeneity"
_TMLR — Accepted by TMLR_

### Review · Reviewer_cUMT · 2024-06-11

**Summary Of Contributions:**

This paper studies the problem of partial personalization for federated learning. The effects of having partial personalization, where the variables are split into local private parameters and global parameters that are shared among the clients, while the clients are stateless, on the accuracy of the model are studied.

**Audience:**

Yes

**Broader Impact Concerns:**

I do not see any ethical implications of the work.

**Claims And Evidence:**

Yes

**Requested Changes:**

It is important to respond to the questions in the previous section. Additionally, here are some minor comments:
- Subsections 1.3-1.5 seem a little out of place. If those are related works, put them under that section, while if they are contributions, then put them under contributions. Having them as just subsections in the introduction is a little confusing.
- There are some typos in the text, please proofread.

**Strengths And Weaknesses:**

The paper is well written and generally easy to follow. The idea is interesting and the work shows that there is some benefit to having partial personalization.
However, in some places, further explanation is required:
- When comparing to the work of Pillutla et al. in the experiments, the comparison seems to be only with the FedAlt. Despite that in the paper by Pillutla et al., FedAlt almost always outperforms FedSim, however, since here the work is using the stateless clients and possibly changing a little in the problem, it would be good to see if FedSim would do better. Moreover, the difference between the FDGG used in this work and FedAlt seems to be very small, so it would be good to understand better why one would prefer to use this method.
- In the experiments, the FDGG method that performs best uses one partitioning scheme or another depending on the dataset. It is important to try to explain why one is preferable to another (or some intuition), what are the possible factors, and how would one decide which to use when using this method.
- In the proof of proposition 1, it is unclear to me how one would pick such a $\theta$ that all $f_m$ do not depend on.
- In section 3.2, on what basis do you pick the values of the parameters in these experiments?

---

> ### Author Response · Authors · 2024-12-16
>
> We thank the reviewer for their feedback and time. Below, we address all the questions and comments raised in the review. The modifications are highlighted in blue color.
>
> >**When comparing to the work of Pillutla et al. in the experiments, the comparison seems to be only with the FedAlt. Despite that in the paper by Pillutla et al., FedAlt almost always outperforms FedSim, however, since here the work is using the stateless clients and possibly changing a little in the problem, it would be good to see if FedSim would do better. Moreover, the difference between the FDGG used in this work and FedAlt seems to be very small, so it would be good to understand better why one would prefer to use this method.**
>
> Of course, we are happy to add the numbers for FedSim to our table (see the last line in the updated Table 2) and add a comment about this in the text (see the second part of Section 3). We do note that FedAlt outperforms FedSim in the stateless setting as well, which was also observed by (Pillutla et al., 2022b), as they make the following claim in their paper: “FedAlt is better than FedSim for stateless devices, although the improvement is smaller.”
>
> >**In the experiments, the FDGG method that performs best uses one partitioning scheme or another depending on the dataset. It is important to try to explain why one is preferable to another (or some intuition), what are the possible factors, and how would one decide which to use when using this method.**
>
> The difference between the numbers for different partitioning schemes is affected by the number of parameters. For instance, on GLDv2, the input layer is 0.01M parameters, the last layer is 1M and the adapter layer is 1.4M. For StackOverflow, in contrast, both input and output layers are 0.8M, while the adapter layer is 0.07M. Therefore, it is not surprising that tuning the adapter layer works great on GLDv2 and it does worse than fine-tuning the last layer on StackOverflow. We also note that we used the same settings as in Pillutla et al. (2022b) for a fair comparison.
>
> >**In the proof of proposition 1, it is unclear to me how one would pick such a $\theta$ that all $f_m$ do not depend on.**
>
> Thanks for pointing out that this detail is unclear, we meant to say that $\theta$ is a dummy variable, for instance, if our function was $f(x_1, x_2) = x_1^2$, then we would use $w=x_1$ and $\theta=x_2$.
>
> >**In section 3.2, on what basis do you pick the values of the parameters in these experiments?**
>
> The stepsizes for all methods are chosen according to theoretical results in corresponding papers. The local stepsize for Scaffold should scale with $1/(L\cdot \tau)$ [Karimireddy et al., 2020], while the global step size is chosen as a numerical constant. In our case, we decided to choose $0.5$ as it was slightly more stable than the step size $1$ used [Karimireddy et al., 2020]. For Local GD the stepsize is scaled with $1/(L\cdot\tau)$ as well due to the results from [Khaled et al., 2020]. Finally, for L2GD we choose $\lambda=0.1$ to have a solution which to have a solution which is in between personalized and non-personalized. The stepsize for L2GD is chosen according to theoretical results in [Hanzely & Richtárik, 2020].
>
> >**Subsections 1.3-1.5 seem a little out of place. If those are related works, put them under that section, while if they are contributions, then put them under contributions. Having them as just subsections in the introduction is a little confusing.**
>
> Thanks for pointing this out, these sections were meant to provide motivation for the issues that we address with our methods. We made this more clear by combining them into one subsection and giving a more explicit explanation for mentioning them.
>
> >**There are some typos in the text, please proofread.**
>
> We have made another pass over the paper to fix typos. Some of the typos that we found and fixed:
> We added missing commas in the proofs (see pages 20, 21, 22, 23)
> We fixed a typo in a derivation on page 23 (after the words “Using (16), we have”)
> We fixed a reference to Section 3 in Appendix E.3
> We replaced “Details” with “details” in the title of Appendix E.4.
> We have not found other typos.
>
> >**Claims And Evidence: No**
>
> Could the reviewer please mention what claims they believe were incorrect or not supported by evidence? We’ll be happy to revise the paper accordingly to fix any issues currently present.

---

> > ### Comment · Reviewer_cUMT · 2024-12-18
> >
> > Thank you for your clarifications. I have no further concerns.

---

### Review · Reviewer_8Vd9 · 2024-07-06

**Summary Of Contributions:**

This paper studies the problem of Partially Personalized Federated Learning, which refers to the Federated Learning scheme where part of the model parameters are tuned in a client-dependent manner. This can be seen as an analog of pre-training and fine-tuning and is a trending paradigm for Federated Learning in recent years [1]. Specifically, this paper improves upon [1] by (1) proposing a stateless optimization scheme where the clients are assumed to be unable to maintain the state of the model and (2) relaxing the assumption on the step size. In addition, this paper also identifies the problem of finding the appropriate split to control the size of shared parameters and private parameters.

[1] Federated learning with partial model personalization, ICML 2022.

**Audience:**

Yes

**Broader Impact Concerns:**

Not applicable.

**Claims And Evidence:**

Yes

**Requested Changes:**

1. Summarize and highlight the contributions of the theoretical analysis, differentiating them between those

2. Highlight how the stateless is achieved in your proposed method.

3. Authors might consider using a separate section to introduce the methodology part.

**Strengths And Weaknesses:**

Strengths:

1. The theoretical analysis in this paper seems solid and comprehensive. However, one suggestion I have is that the author could consider summarizing their theoretical contributions into takeaway messages so that the audience can directly understand how it contributes to the community compared with existing works.

2. This paper identifies an interesting problem - over-personalization, which can be seen as finding the right amount of parameter split between the shared parameters and private parameters. This question seems intuitive, and this paper proposes theoretical insights to guide this process.

Weaknesses:

1. Authors should highlight how stateless property is achieved using their proposed method, but this point is not clear in the current version of the paper.

2. From a high level, the difference between this work and [1] is identified in Table 1; however, in detail (e.g., methodology level), how does the proposed method differ from the one proposed in [1]?

3. Is the studied overpersonalization regime client-dependent? Intuitively, based on different amounts of local data, as well as the complexity of the distribution, the number of private parameters should be client-dependent. Is this point reflected in your analysis or algorithm?

[1] Federated learning with partial model personalization, ICML 2022.

---

> ### Author Response · Authors · 2024-12-16
>
> Thank you for providing us with feedback on our work and commending our theoretical analysis!
>
> > **one suggestion I have is that the author could consider summarizing their theoretical contributions into takeaway messages**
>
> We thank the reviewer for this suggestion, we’ve added a few takeaway points in the Conclusions section of the newly submitted version.
>
> > **Authors should highlight how stateless property is achieved using their proposed method**
>
> It should have been indeed discussed in the submission, we totally missed it. We’ve added a paragraph about our method’s properties in Section 2.1 to fix this.
>
> > **how does the proposed method differ from the one proposed in [1]?**
>
> We added an extended discussion on this in the updated paper in Section 3, but let us provide a response here as well. From the methodological perspective, the approach in [1] seems to take the gradient-based methods and study if they would solve the problem. Our approach, in contrast, is to notice that the objective is similar to bilevel optimization with the restriction that we cannot store auxiliary variables due to the stateless nature of the clients. Therefore, the method that we proposed tries to first find appropriate private parameters $w_m(\theta)$ and only then update $\theta$, whereas the FedSim method in [1] updates $\theta$ on each worker without first trying to see if a client’s local objective can be minimized by updating only $w_m$. Their FedAlt method is similar to our FFGG in case they set $\tau_v=1$ and if we use gradient descent as the local solver when optimizing $w_m$. This is not, however, supported by their github implementation, which requires the number of iterations with respect to $\theta$ and $w_m$ to be the same. While it's not explicitly stated in their paper, it appears they did all experiments using this approach, which is why we see different numerical performances. Finally, our approach is slightly more flexible, since we can use any other solver, e.g., Adam, to do the task of fine-tuning.
>
> > **the number of private parameters should be client-dependent. Is this point reflected in your analysis or algorithm?**
>
> Indeed, it makes sense for the number of parameters updated by a client to depend on how different its data distribution is from the rest of the clients. Our problem formulation takes this into account by allowing them to have an arbitrary number of local parameters, which in practice can be implemented by changing the rank in LoRA or changing the number of adapter/LoRA layers. A limitation of our work, which we acknowledge in the text, is that we do not have a solution to how one should decide on the number of parameters, but some recent work (Yang et al, 2023) has addressed this issue.
>
> > **Requested changes**
>
> We have changed the paper as per your request. Please let us know if any other changes are required.
>
> [Yang et al, 2023] X. Yang, W. Huang, M. Ye, “Dynamic Personalized Federated Learning with Adaptive Differential Privacy”, 2023

---

### Review · Reviewer_XKnK · 2024-11-29

**Summary Of Contributions:**

The paper introduces a partially personalized formulation of Federated Learning. The paper claims that the data heterogeneity slowdown can be completely eradicated via over-personalization. The paper proves that, in contrast to standard FL, asynchronous training with partial personalization converges precisely, and partial personalization can be made Byzantine robust.

The proposed algorithm divides the model parameters into global and local (personalized) set. The training includes three main steps:
1. Non-personalized federated pre-training: Here the global set of parameters are trained using FedAvg without any personalization (the local set of parameters are frozen in this stage).
2. Partially personalized federated training: In this stage, both the global and local parameters of the model are trained using Algorithm 1. Here, the following two steps are repeated until convergence -  a) The local parameters are randomly initialized and fine-tuned based on the local objective for $\tau$ iterations, and b) perform one FedAvg step on the global parameters with number of local steps set to 1.
3. Final fine-tuning: This step involves only fine-tuning the local (personalized) parameters for each client.

An extension to Algorithm 1 to incorporate local steps with respect to global and local parameters into the training loop is presented as Algorithm 4.

**Audience:**

Yes

**Claims And Evidence:**

Yes

**Requested Changes:**

1. Please add a separate section for methodology.
2. Add a separate limitation section discussing the overheads of the proposed algorithm.
3. Include some experimental results to support the claims on data heterogeneity, asynchronous training and byzantine robustness.
4. Since FedAvg (no personalization) and Local GD (only personalization) are the two extremes of the setup, it is useful to add Local GD results in Table 2.

Minor changes:
1. The reference in point 2 of section E.3 is missing (indicated as ??).

**Strengths And Weaknesses:**

Strengths:
1. The paper addresses an important problem of partially personalized FL to address data heterogeneity.
2. The paper proposes various modification to the proposed algorithm to adapt it to asynchronous learning and byzantine robustness.
3. The clients are stateless by default.
4. The paper presents a comprehensive theoretical analysis.

Weaknesses/Questions:
1. The paper is hard to follow. Please add a separate section for methodology and clearly describe the algorithms and proposed modifications/variations of algorithms before moving into the theory.
2. The proposed algorithm has computation and latency overheads. For example, in algorithm 1, step 4 requires multiple iterations before 1 iteration of FedAvg which increases the compute and latency. Similarly Algorithm 4 requires multiple backward steps. The paper doesn't discuss any limitations/over-heads of the proposed method.
3. The paper has limited experimental results. Not all the claims are backed up by the experimental results.
4. In table 1, why is FedProx not listed under "Handles data heterogeneity"?
5. What is the type of and level of data heterogeneity (if any) used in the experiments (Table 2)?
6. Does the results in table 2 use Algorithm 1 or 4?
7. In figure 1, $\tau$ is related to the local steps to tune local parameters and this is a hyper-parameter that is specific to FFGG. So, why do the curves wrt to other algorithms (scaffold, local GD etc) are different across graphs b, c, and d? Please provide some clarity on how to interpret the figure.

---

> ### Author Response · Authors · 2024-12-16
>
> We thank the reviewer for their feedback and time. We appreciate that you identified the studied problem as important and that you found our theoretical analysis to be comprehensive. Below, we address all the questions and comments raised in the review. The modifications are highlighted in blue color.
>
> >**The paper is hard to follow. Please add a separate section for methodology and clearly describe the algorithms and proposed modifications/variations of algorithms before moving into the theory.**
> >**Please add a separate section for methodology.**
>
> Following the reviewer’s suggestion, we separated the algorithm’s descriptions from the theoretical convergence results. If you have any further comments on how we can improve readability, we will be happy to incorporate them. We made a pass over the paper to clarify places that we thought might look confusing (the changed parts are highlighted in blue). We are also planning to go through the whole paper again to improve writing for the final version of our work.
>
> >**The proposed algorithm has computation and latency overheads. For example, in algorithm 1, step 4 requires multiple iterations before 1 iteration of FedAvg which increases the compute and latency. Similarly Algorithm 4 requires multiple backward steps. The paper doesn't discuss any limitations/over-heads of the proposed method.**
> >**Add a separate limitation section discussing the overheads of the proposed algorithm.**
>
> We thank the reviewer for pointing out this limitation. We added Section 6 that provides a discussion of the limitations of our work. We also point out that the considered methods do not have computational/latency overhead in comparison to standard FedAvg: in FedAvg, the clients also perform local steps. On top of that, our FFGG has slightly cheaper iterations than FedAvg since FFGG with Local-GD subsolver computes multiple gradients with respect to personalized parameters and one gradient with respect to global parameters, while FedAvg requires the clients to compute multiple gradients with respect to **all** parameters.
>
> >**The paper has limited experimental results. Not all the claims are backed up by the experimental results.**
> >**Include some experimental results to support the claims on data heterogeneity, asynchronous training and byzantine robustness.**
>
> Following the reviewer’s suggestion, we conducted additional numerical experiments supporting the findings of the paper. In Appendix E.1, you can find our new experiments on the impact of data heterogeneity, where we show that our method is way less impacted by the data heterogeneity than the other methods. We are working on adding additional results on the asynchronous training and Byzantine robustness, they will be available a little bit later.
>
> >**In table 1, why is FedProx not listed under "Handles data heterogeneity"?**
>
> As we write in the caption of Table 1, we say that “a method handles heterogeneous data if its complexity does not depend on any heterogeneity constant.” The complexity of FedProx does depend on the data heterogeneity. For example, for linear losses, FedProx is equivalent to FedAvg, which is known to slow down under high data heterogeneity.
>
> >**What is the type of and level of data heterogeneity (if any) used in the experiments (Table 2)?**
>
> The split of data among clients is described for each particular task in Appendix E (see paragraphs “Client distributions”). We did not measure the heterogeneity level for these tasks, but since in all of these examples each client corresponds to an individual user the level of heterogeneity in these experiments is realistic.
>
> >**Does the results in table 2 use Algorithm 1 or 4?**
>
> We use Algorithm 4 in these experiments.
>
> >**In figure 1, $\tau$ is related to the local steps to tune local parameters and this is a hyper-parameter that is specific to FFGG. So, why do the curves wrt to other algorithms (scaffold, local GD etc) are different across graphs b, c, and d? Please provide some clarity on how to interpret the figure.**
>
> Scaffold, Local GD and L2GD have their own inner loops with a flexible number of local steps, which we denote for convenience as $\tau$ for all algorithms. The notation might be different in the original papers, so we understand why it might be confusing, we added a clarification in the figure description.
>
> >**Since FedAvg (no personalization) and Local GD (only personalization) are the two extremes of the setup, it is useful to add Local GD results in Table 2.**
>
> If we understand the reviewer correctly, by Local GD the reviewer means the method where clients simply optimize their local loss functions with GD without communicating at all. This is indeed a good idea to showcase the utility of collaboration between the clients, so we have added the corresponding results to Table 2. Unsurprisingly, the limited local data available on the clients leads to poor performance of the models trained without collaboration.

---

### Review · Reviewer_attF · 2024-12-22

**Summary Of Contributions:**

This paper introduces a Partially Personalized Federated Learning framework to address data heterogeneity in FL by splitting model parameters into global and local components. It claims that an optimal split allows for global parameters enabling perfect data fitting at each client, termed as "overpersonalized." The proposed algorithm shows benefits in various FL settings, including local steps, asynchronous training, and Byzantine-robust training. The paper also discusses the practical applications of partial personalization in representation learning and heterogeneous environments and analyzes the method's convergence and robustness against Byzantine attacks under data heterogeneity.

**Audience:**

No

**Broader Impact Concerns:**

Not applicable.

**Claims And Evidence:**

Yes

**Requested Changes:**

-	Introduce a dedicated methodology section that clearly outlines the algorithms and their step-by-step operations within the Partially Personalized Federated Learning framework. Include pseudocode or flowcharts for better understanding and use visual aids to elucidate complex concepts. Ensure that any algorithmic modifications are thoroughly explained, with emphasis on their distinctions from existing methods and the reasoning behind them.
-	Broaden the experimental section to include a more comprehensive comparative analysis against a wider array of state-of-the-art Federated Learning methods. This expansion will provide a stronger empirical foundation for the paper's claims and demonstrate the practical superiority of the proposed algorithm.
-	Address the critical gap in the optimization of the parameter split between global and local components. Provide a detailed analysis or propose a methodology for determining the optimal split, which is essential for balancing model complexity, client-side storage requirements, and the effectiveness of personalization in the framework.
-	The manuscript should include a clear and concise comparative analysis that explicitly differentiates the proposed method from existing work in the field. This section should articulate the novel aspects of the approach and demonstrate its advantages, providing specific examples of how it improves upon or addresses limitations of previous methods.

**Strengths And Weaknesses:**

Strength:

-	This paper proposes a novel approach to FL by introducing partially personalized parameters, which includes both globally shared and client-specific local parameters, to address the challenge of data heterogeneity across different clients.
-	It provides theoretical proof that under the right parameter split, it is possible to find global parameters allowing each client to fit their data perfectly, an idea the authors refer to as "overpersonalized," which is a meaningful contribution to the field of personalized FL.
-	It analyzes the algorithm's performance under asynchronous training conditions and in the presence of Byzantine attacks, asserting that the method can converge under arbitrary delays and remain robust even with heterogeneous data, which is a significant advancement for the security and reliability of FL systems.

Weakness:

-	The manuscript urgently needs a dedicated methodology section for clarity and readability. This section should explicitly describe the algorithms in the Partially Personalized Federated Learning framework with step-by-step explanations, pseudocode, or flowcharts. Any modifications to existing algorithms must be detailed, emphasizing their differences and rationale. Visual aids (e.g., figures) can be incorporated to clarify complex ideas, and key points should be reiterated for emphasis. Theoretical discussions should follow only after establishing this practical foundation.
-	While the paper presents a theoretically sound approach, its experimental validation is insufficient. The comparative analysis lacks breadth, falling short of evaluating the proposed algorithm against a wide range of state-of-the-art FL methods. To strengthen its claims, the paper should include a comprehensive set of benchmarks and demonstrate superiority in practical scenarios.
-	The concept of parameter splitting between global and local components is introduced but lacks a detailed explanation. The choice of parameter split is critical for performance, affecting model complexity, client-side storage, and personalization effectiveness, yet the paper offers no detailed methodology for determining the optimal split. Addressing this gap with a more nuanced optimization approach would significantly enhance the framework's practical utility.
-	Elaborate on how the proposed Partially Personalized Federated Learning approach differs from and improves/surpasses existing methods in a comparative manner, highlighting the unique contributions and improvements over the state-of-the-art.

---

> ### Author Response · Authors · 2025-01-06
>
> We appreciate the reviewer's thoughtful comments. We appreciate that you called our analysis “a significant advancement for the security and reliability of FL systems”. We understand your point about the weaknesses of our work, although we respectfully disagree with some concerns:
> > This section should explicitly describe the algorithms in the Partially Personalized Federated Learning framework with step-by-step explanations, pseudocode, or flowcharts.
>
> We provided the algorithms studied in our work on pages 6 (main algorithms), 8 (asynchronous and local), and 27 (Byzantine-robust formulation). We provided the main differences to related work in Section 3, which was added recently following the comments from other reviewers, so we realize the reviewer might not have seen it if they had started reviewing our paper earlier.
> > Theoretical discussions should follow only after establishing this practical foundation.
>
> Our paper is primarily theoretical, proving the fundamental properties of partially personalized FL. We believe that the theoretical foundation must precede implementation details, as it establishes what's achievable in this framework, and it is the default way in papers on the theory of federated learning algorithms.
> > The comparative analysis lacks breadth, falling short of evaluating the proposed algorithm against a wide range of state-of-the-art FL methods.
>
> Could the reviewer be more specific in what aspect our work fails to compare to the state of the art? We have performed experiments on all benchmarks used by Pillutla et al., which to the best of our knowledge is state of the art. You can find comparisons to FedAvg, FedAlt, FedSim, and Local Training in our Table 2. On top of that, we have additional results in Figures 1 and 2 comparing to Scaffold, Local GD, and L2GD. We can add results for Ditto and pFedMe to Table 2, but it is important to take into account that they personalize the whole model rather than a subset of parameters and can be less practical in real-world scenarios. Finally, we note that the goal of our experiments is to *demonstrate the theoretical properties* of our methods, our main contribution remains to be theoretical.
> > the paper offers no detailed methodology for determining the optimal split.
>
> In our experimental evaluation in Section 5.3, we compare 3 approaches: fine-tuning the input layer, the output layer, and using Adapter layers. As we observe, the low-rank adapter layers perform the best on the Federated Extended MNIST, and Google Landmarks Dataset v2. On StackOverflow, using Adapter is almost as good as fine-tuning the last layer. We find this to be strong evidence that low-rank fine-tuning, which is also very popular in practice in other settings, is the best practical approach to partial personalization. We also note that the question of optimally placing the fine-tuned parameters is an open problem even outside of the context of federated learning, see for example Nowak et al. “Towards Optimal Adapter Placement for Efficient Transfer Learning”.
> > Elaborate on how the proposed Partially Personalized Federated Learning approach differs from and improves/surpasses existing methods in a comparative manner, highlighting the unique contributions and improvements over the state-of-the-art.
>
> We have provided a comparison in Table 1, where we emphasized that a key feature of our work is supporting stateless clients, heterogeneous data, as well as showing the theoretical benefit of local training. In Section 3, we also explain the technical differences to the work of Pillutla et al. We will clarify these points in revision while maintaining our paper's theoretical focus.
> Would the reviewer prefer we add a subsection before the theoretical development to point to the empirical evaluations and conclusions?

---

### Decision · Action_Editor_756y · 2025-01-06

**Recommendation:** Accept as is

**Comment:**

This paper analyzes an approach for stateless personalized federated learning by considering modeling via a combination of global and local-only parameters. The authors analyze their approach in several scenarios, including communication-efficient training regimes such as local-updating & asynchronous training scenarios, as well as in the presence of malicious clients. The reviewers provided significant suggestions to improve the presentation of the work and better position the work with respect to prior art in personalized FL, which the authors have incorporated in their revision. Although some uncertainty remains regarding the empirical improvements of this approach relative to existing methods, reviewers were nonetheless impressed with the theoretical insights in the work and uniformly recommended acceptance of the paper.

**Audience:**

This work explores a method for personalized federated learning, which is of relevance to the TMLR community.

**Claims And Evidence:**

Although reviewers thought the work could be strengthened by demonstrating the empirical success of the method across a wider range of benchmarks and baseline methods, all agreed that the method and theoretical analysis was sound and a useful contribution to the field.